# Machine learning for automated electrical penetration graph analysis of aphid feeding behavior: Accelerating research on insect-plant interactions

**Quang Dung Dinh**[1], **Daniel Kunk**[2,3], **Truong Son Hy**[5]*,
**Vamsi Nalam**[2,3]*, **Phuong D. Dao**[2,4]

**1** Institut Galilée, Universite Sorbonne Paris Nord, Villetaneuse, Paris, France, **2** Department of Agricultural Biology, Colorado State University, Fort Collins, Colorado, United States of America, **3** Department of Cell and Molecular Biology, Colorado State University, Fort Collins, Colorado, United States of America, **4** Graduate Degree Program in Ecology, Colorado State University, Fort Collins, Colorado, United States of America, **5** Department of Computer Science, The University of Alabama at Birmingham, Birmingham, Alabama, United States of America

* thy@uab.edu (TSH); Vamsi.Nalam@colostate.edu (VN)

**Data availability statement:** All code written in support of this publication is publicly available at https://github.com/HySonLab/ML4Insects. All data used in the publication is from previously published manuscripts and the relevant references have been provided in the text.

**Funding:** The work was supported by start-up funds provided to Drs. Nalam and Dao from Colorado State University.

## Abstract

The electrical penetration graph (EPG) is a well-known technique that provides insights into the feeding behavior of insects with piercing-sucking mouthparts, mostly hemipterans. Since its inception in the 1960s, EPG has become indispensable in studying plant-insect interactions, revealing critical information about host plant selection, plant resistance, virus transmission, and responses to environmental factors. By integrating the plant and insect into an electrical circuit, EPG allows researchers to identify specific feeding behaviors based on their distinctive waveform patterns. However, the traditional manual analysis of EPG waveform data is time-consuming and labor-intensive, limiting research throughput. This study presents a novel Machine Learning (ML) approach to automate the annotation of EPG signals. We rigorously evaluated six diverse ML models, including neural networks, tree-based models, and logistic regression, using an extensive dataset from multiple aphid feeding experiments. Our results demonstrate that a Residual Network (ResNet) architecture achieved the highest overall waveform classification accuracy of 96.8% and highest segmentation overlap rate of 84.4%, highlighting the potential of ML for accurate and efficient EPG analysis. This automated approach promises to accelerate research in this field significantly and broaden insights into insect-plant interactions, showcasing the power of computational techniques for insect biological research. The source code for all experiments conducted within this study is publicly available at https://github.com/HySonLab/ML4Insects.

## Introduction

Aphids (order Hemiptera) are major agricultural pests that feed on plant phloem sap using their needle-like mouthparts. Their rapid asexual reproduction and high population densities

**Competing interests:** The authors have declared that no competing interests exist.

can lead to significant crop damage. Copious feeding by aphids depletes essential nutrients from plants and introduces potentially toxic saliva and plant viruses, further compromising crop quality and yield. In severe cases, aphid infestations can result in hundreds of millions of dollars in economic losses, underscoring the importance of understanding and managing these pests [1]. The electrical penetration graph (EPG) technique, introduced in 1964 by Donald McLean and Marvin Kinsey, provided researchers with a tool to better understand aphid feeding behavior. The initial EPG technique involved creating an electrical circuit by wiring the aphid and the plant, then recording the resulting voltage changes once the aphid closes the circuit by penetrating the plant tissue [2]. Throughout the 20th century, entomologists enhanced this technique by analyzing the behavioral mechanisms behind EPG waveforms. This was achieved by performing stylectomy followed by dissection of the plant tissue to determine the approximate location of the stylet at the time the behavior was recorded [3]. This research further allowed for the understanding of the underlying electrical component of each behavioral waveform, and through the invention of the direct current (DC) EPG system, it allowed researchers to record behaviors originating from both plant-insect resistance (Ri) and those originating from the electromotive force (emf) [4,5]. Today, the EPG technique stands as a leading method for studying hemipteran feeding behavior [6,7] .

Currently, commercially available DC-EPG systems provide the necessary hardware and software for acquiring EPG recordings [8]. To set up and create the circuit, aphids are restrained, and a gold wire is glued to their abdomen using conductive paint. The gold wire is typically glued to a copper wire that is soldered onto a brass nail, which can be inserted into a specially designed housing attached to the EPG circuit. For the plant, a brass electrode is inserted into the soil and connected to the EPG circuit. The analog signal is converted to a digital signal in GIGA systems and recorded on a computer using the Stylet+ data acquisition software (.DAQ file and .DX extension for hour-by-hour recordings). After recording feeding behaviors, the user manually annotates the waveforms by visual inspection, creating a file with columns for the start time, the specific behavior, and the voltage value at the start of the behavior. Various parameters associated with feeding behaviors are then manually calculated from these annotation files or using publicly available Excel workbooks [9].

However, the throughput of EPG analysis is greatly constrained because it demands a time-consuming manual review process and the expertise of an experienced observer to annotate the signals accurately. During each annotation session, the observer analyzes data on a second-by-second basis, which can take up to 30 minutes (for an 8 h recording), depending on the complexity of the observed patterns and the length of the recording. Additionally, recordings with ambiguous waveforms, excessive noise, or numerous individual events can significantly prolong the annotation process. Thus, an automated annotation pipeline will greatly streamline data analysis. Previous attempts at automation, such as Aphid Auto EPG [10], had limited classification ability due to its basic discrimination rules based on waveform amplitudes. Similarly, Adasme-Carreño et al. (2015) developed the Assisted Analysis of EPG (A2EPG) program to automate the annotation of EPG signals [11]. However, the criteria for distinguishing waveforms was overly simplistic, leading to the misclassification of waveforms.

Researchers have explored ML methods for EPG signals due to their versatility with diverse data types and reliable predictive performance. In 2018, Wu et al. [12] (2018) used Decision Tree (DT) for classifying samples of EPG waveform obtained from the green peach aphid (*Myzus persicae*) feeding on tobaco host plants (Zhongyan No. 1 variety), reporting a mean classification accuracy of 91.43% over 4 experiments. On the same dataset, Xing et al. (2023) [13] proposed using Extreme Kernel Machine (EKM) instead of DT improviing the prediction accuracy to 94.47%. A Random Forest model was introduced to characterize the behaviors of Asian citrus psyllids (*Diaporina citri*) on nine citrus genotypes [14], achieving an

overall accuracy score of 97.4%, with the Hidden Markov Model further revealing previously unknown feeding patterns. However, these studies were limited in scale, necessitating further validation of larger and more diverse datasets to assess the broader applicability of these methods. In recent years, Convolutional Neural Networks (CNNs) have shown great promise in insect behavior monitoring [15–19], with their ability to automatically learn informative features from raw data eliminating the need for manual feature engineering. Furthermore, CNN variants have proven effective in segmenting biological time series [20–23], showcasing their potential in EPG analysis.

In this study, we aim to 1) evaluate the performance of well-established ML models for characterizing and annotating EPG signals, 2) evaluate different feature extraction methods on model performance, and 3) develop a ready-to-use pipeline for automatic waveform annotation for users who may not be familiar with ML tools. Our study builds on previous work by evaluating various ML models, including *Fully Convolutional Network* (1DCNN), *2D Convolutional Neural Network* (2DCNN), *Residual Network* (ResNet), *Extreme Gradient Boosting classifier* (XGB), *Random Forest* (RF), and *Logistic Regression* (LR). A Python package was developed based on well-established libraries such as Scikit-learn and PyTorch, providing an end-to-end automatic annotation process for reproducibility and widespread use.

## Materials and methods

### EPG signal annotation and characterization

**Aphid's feeding behavior.** Waveforms within the EPG signals are classified into seven categories: non-penetrating (NP), pathway (C), potential drop (pd), phloem salivation (E1), phloem-feeding (E2), xylem feeding (G), and derailed stylet phase (F), depending on the pattern observed from both the time and frequency domain. During the NP phase, the insect has yet to insert its mouthpart into the host plant, so the waveform is usually almost a straight line. Pathway phase (C) is when the stylet is inserted into the plant tissue and is present in the epidermal and mesophyll tissues. During C, brief intracellular punctures by stylet tips occur, referred to as pd or potential drops. It is during C that aphids make decisions regarding host acceptance or rejection. The E1 and E2 phases are sometimes combined and called the sieve element phase (SEP). During SEP, the stylet tips are within phloem sieve elements. During E1, the aphid actively salivates while maintaining stylet position within the sieve elements. The E2 phase signifies active phloem sap ingestion, with the stylet remaining in the sieve element. In contrast, the G waveform indicates xylem sap ingestion, characterized by the stylet positioned in the xylem. Lastly, the F waveform represents derailed stylets, where the aphid encounters mechanical difficulties while probing host tissue. In this phase, despite being within host tissue, the aphid does not engage in feeding behaviors due to stylet malfunctions. It is important to note that these behaviors are sequential. For instance, the aphid must enter C before engaging in phloem or xylem feeding (E1, E2, or G), and the pd pattern is only observed during C. Examples of waveforms of these behaviors are shown in (Fig 1).

**Workflow.** We propose an end-to-end pipeline following a sliding window approach to automate the EPG signal annotation process. We attempt to create a complete partition of an input recording by making predictions on the component chunks of the signal. Our pipeline consists of three main steps: initial segmentation, waveform sample classification, and label aggregation (Fig 2). Specifically, the input signal is initially divided into consecutive segments of fixed-length $d$ with a step size $s$. By default, we set $d = 1024$ (corresponding to a 10.24s segment) and $s = d$ to obtain a non-overlap segment. Then, the segments of signals are transformed using Fourier or Wavelet transformation into valuable features that characterize the

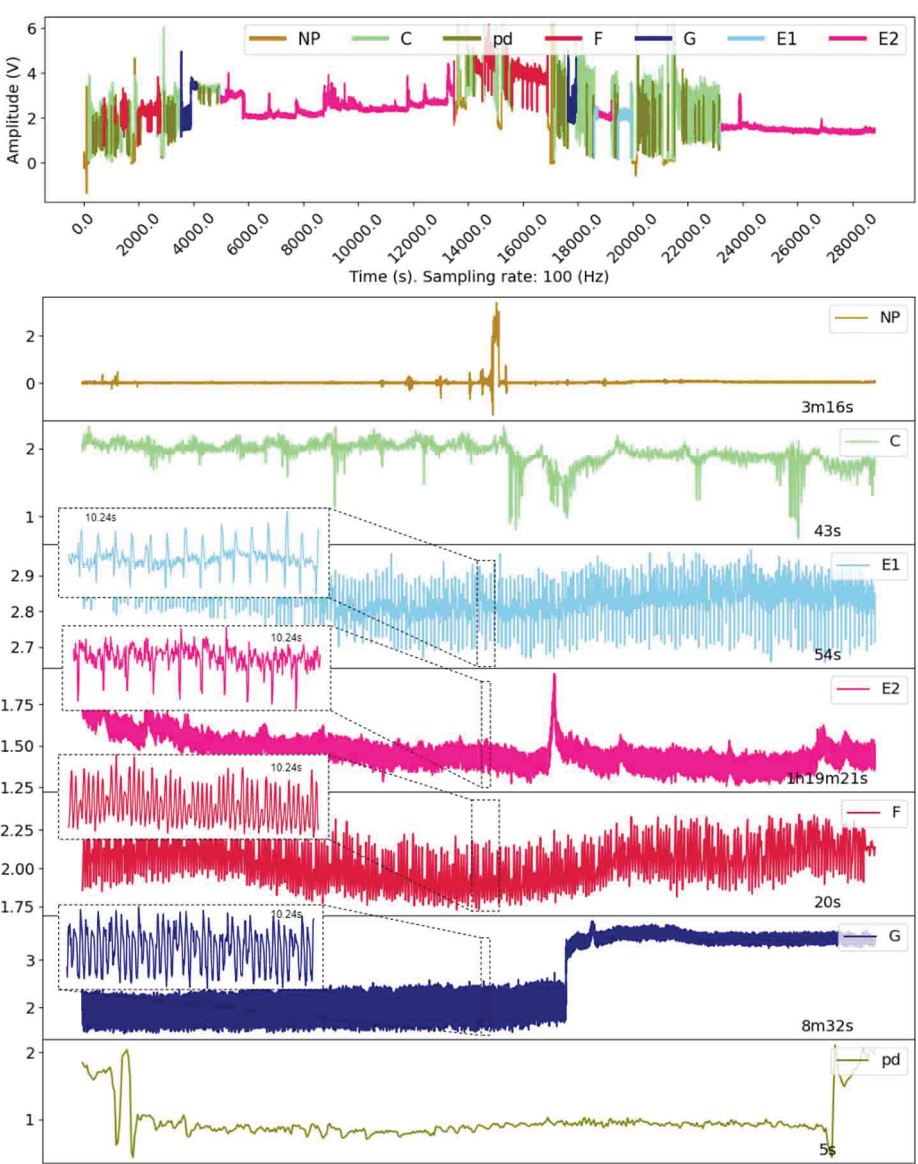

**Fig 1. Representative waveforms observed during aphid feeding.** The top panel shows a full EPG recording of 8 h (28,800 s). The following panels display segmented waveforms corresponding to seven distinct aphid feeding behaviors. The non-probing phase (NP) is typically flat (0 mV) with occasional noisy peaks. The pathway phase (C) includes a range of complex behaviors, with the pd phase occuring within it. The sieve element phase consists of E1 and E2, while the derailed stylet (F) and xylem phase (G) are characterized by unique waveform shapes and frequencies/periods. Note that waveform patterns may vary by aphid species, host plant(s), and environmental conditions of the experiment.

signal's behavior. The Data Processing section discusses this step in greater detail. In the second step, trained ML models will classify the segments using the transformed data as input by generating a probability distribution illustrating the likelihood that a segment belongs to a specific waveform. Each segment will be assigned a label with the highest probability. The following sections discuss the details of the models studied in this work. Lastly, the predicted labels will be concatenated to create a complete segmentation of the input recording. To do

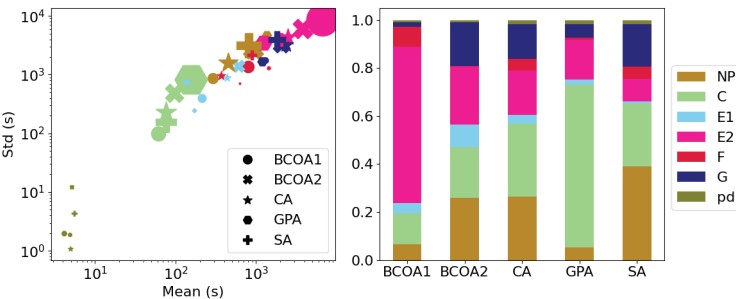

**Fig 2. Automated EPG signal annotation and characterization pipeline.** End-to-end process of automating EPG signal analysis. Input recordings are divided into fixed-length segments, which undergo feature extraction. Machine Learning (ML) models learn from the extracted features or raw signals to predict behaviors. Each segment is then assigned to a class of the highest probability. Finally, the predicted labels are concatenated to form a complete annotated recording.

this, each time step is given a predicted label according to that of the segment containing it; consequently, each segment yields *d* labels for *d* time steps. A complete predicted waveform is thus composed by concatenating such sequences, from which one may determine the predicted locations of each waveform. For example, assume that an EPG recording is 8 hours long, recorded at a sampling frequency of 100Hz (i.e., 100 time steps per second). A waveform sample located between 0s and 10.23s, assigned with label NP, will generate a sequence of 1024 identical labels (NP, NP, ..., NP). Concatenating these sequences of predictions will thus result in 2,880,000 labels for the input recording.

**Classification models.** *Random Forest (RF)*: A well-known ML algorithm used for both classification and regression tasks. An RF consists of multiple Decision Trees (DTs) trained on subdatasets created by bootstrapping from random subsets of features. The time series adaptation of RF [24,25] involves segmenting the input time series into intervals and extracting various features from these segments. Experimental studies show that using RF with simple features such as mean, standard deviation, and slope is computationally efficient and outperforms strong competitors such as 1-NN classifiers with dynamic time-warping distance.

*Extreme Gradient Boosting classifier (XGB)*: A very fast and accurate state-of-the-art tree-based model built under the Gradient Boosting framework. The algorithm operates by sequentially adding weak learners to the ensemble, with each new learner focusing on correcting the errors made by the existing ones while using a gradient descent optimization technique to minimize a predefined loss function during training. In [26], four state-of-the-art gradient boosting-based algorithms, including XGBoost, LGBM, GBM, and CatBoost, are benchmarked based on 12 datasets of different sizes and contexts. The results showed that all models achieved state-of-the-art performance, with LGBM ranking first. However, it is crucial to note that extensive fine-tuning was essential to attain this level of performance.

*Logistic Regression (LogReg)*: A popular and widely used classification algorithm owing to its simplicity, interpretability, and effectiveness for binary classification tasks. Moreover, this method can be extended to multi-class classification problems through one-vs-all or softmax regression methods.

*Convolutional neural networks (CNNs)*: A type of neural network algorithm that learns characteristics of the input data by optimizing the *kernels*. In CNNs, learning units are arranged in layers, and the heart of CNNs is the *matrix convolution* operator used at the *convolutional layers*. At each convolutional layer, the input convolves with the kernels in order

to produce *feature maps*. The kernel allows the network to concentrate on specific parts of the input sequentially, thereby enabling CNNs to minimize the parameters per layer and promote a more extensive network architecture. This feature makes CNNs particularly effective at tasks where focus on local patterns is required, as in the case of image and time-series data.

## Data

The ML models were benchmarked on datasets collected from four aphid species: *Rhopalosiphum padi* (bird cherry-oat aphid; BCOA1, BCOA2) *Myzus persicae* (green peach aphid; GPA), *Phorodon cannabis* (cannabis aphid; CA) and *Aphis glycines* (Soybean aphid; SA) that were obtained from the *Stylet+* application [8]. The datasets include a set of signal data in ASCII format and a text file that provides corresponding ground-truth annotations. Each EPG recording consists of a single signal channel that spans 8 hours at a sampling rate of 100Hz. The components of each dataset are described in (Fig 3). Typically, the patterns from the pathwave phase (C) and sieve element phase (E1, E2) account for a major proportion of the total recording time. In two datasets, GPA and BCOA2, there were virtually no F and G waveforms, which partly explains the difficulty in distinguishing between these waveforms and the others. The potential drop (pd) waveform tends to appear frequently but over a short time, so they account for a negligible proportion of the data distribution.

**BCOA1**. This dataset is derived from an experiment on the bird cherry-oat aphid (*R. padi*) feeding on wheat (*Triticum aestivum* var. Chinese Spring). Aphid feeding was recorded for 8 h over a 24-hour period during the morning, afternoon, and night under two different light conditions: a photoperiod of 16 h light/8 h dark and continuous 24-hour light. The aim of the experiment is to understand how the time of day and light-dark cycles impact feeding behavior. Aphids were allowed to feed from the 2nd leaf of seedlings at Zadoks stage Z1.2 [27]. The characteristics of their feeding waveforms exhibited minimal variation across the different experimental conditions.

**BCOA2** [28]. This dataset is derived from an experiment conducted on the bird cherry-oat aphid (*R. padi*) feeding on Wheat (*Triticum aestivum* var. Coker 9553, AgriPro®). Aphid feeding was recorded at the beginning of the day and the beginning of the night to understand the influence of the time of day on aphid feeding behavior. Aphids and experimental plants were reared in a 16 h light and 8 h dark photoperiod at 24 ± 1 °C. Experiments were conducted on seedlings at Zadoks stage Z1.2.

**CA** [29]. This dataset comes from an experiment conducted on the cannabis aphid (*P. cannabis*) feeding either on its primary host, Hemp (*Cannabis sativa* var. Elite, New West Genetics) or a secondary host, potato (*Solanum tuberosum* var. CO07015-4RU). The aim of this experiment was to understand the differences in the feeding behavior of viruliferous and non-viruliferous aphids on hemp and potato to determine the potential of the cannabis aphid to vector potato virus Y to these species. Experimental plants were reared in a greenhouse in a 16 h light/8 h dark photoperiod at 21 ± 1 °C/16 ± 1 °C, and two three-week-old hemp and potato plants were used in the experiments. Aphids performed poorly on potatoes, probably due to the difference in host plant physiology and cellular anatomy, and waveform characteristics between aphids feeding on potatoes and hemp plants are likely not identical.

**SA** [30]. This dataset is from the soybean aphid (*A. glycines*) feeding on two genotypes of *Glycine max* (soybean). The experiment compared aphid feeding behavior on the susceptible cultivar, Williams 82, and the resistant cultivar PI 567301B, which contains the *Rag5* gene, both on whole plants and detached leaves. The aim was to investigate the origins of *Rag5*-mediated resistance. Aphids and plants were reared under a 16 h light/8 h dark photoperiod at 24 ± 1 °C, with plants at the V1 growth stage used for all recordings. Since experiments were

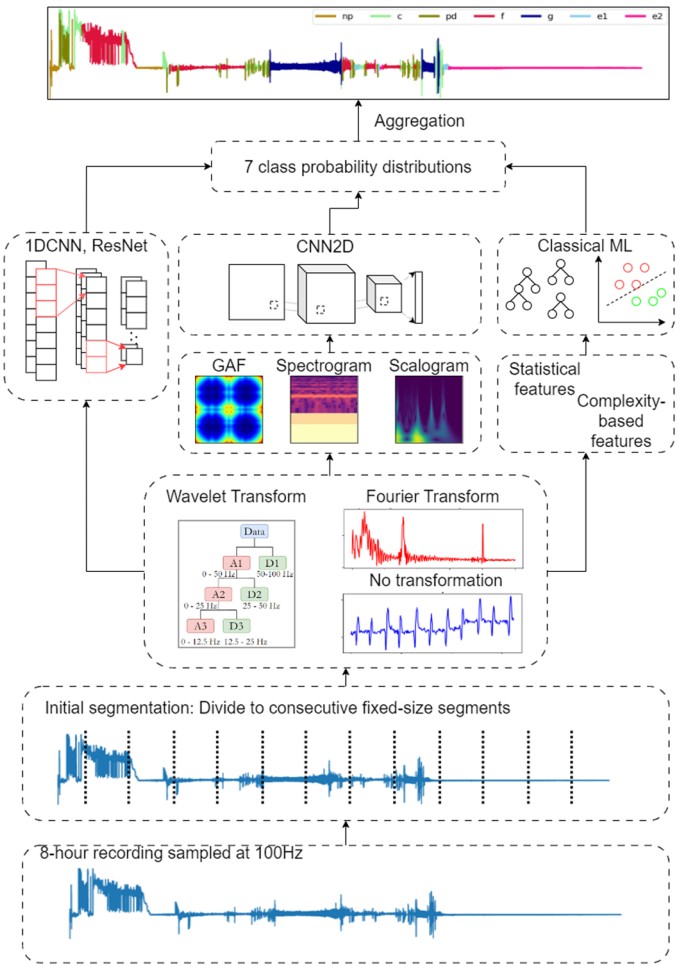

**Fig 3. Details of the tested EPG dataset.** The label distribution (left) versus the mean and standard deviation of each label (right). The size of each marker is proportional to the ratio of the corresponding label in the dataset.

conducted on both whole plants and detached leaves, waveform characteristics may differ across treatments.

**GPA** [31]. This dataset is from experiments with green peach aphid (*M. persicae*) feeding on Arabidopsis. The aim of the experiment is to understand the relationship between plant autophagy and insect feeding behavior and performance. Aphid colonies and experimental plants were reared in a 16 h light and 8 h dark photoperiod at 22 ± 3 °C. Aphids were allowed to feed on one-month-old wildtype Col-0 plants or autophagy mutants *atg5.1* and *atg7.2*. Aphids showed reduced feeding on mutant *atg7.2*; however, waveform characteristics remained consistent across all experimental treatments.

These datasets are summarized in Table 1 for reference.

## Data processing

**Preprocessing. Data splitting.** For each dataset, 90% the total number of recordings (referred to as Subset 1) were used to train ML models and evaluate their classification performance. Subset 1 recordings were divided into fixed-length segments, creating a database

**Table 1. Description of the studied datasets.** The table gives metadata about each dataset, where *n* stands for the total number of 8-hour recordings.

| Dataset | n | Species | Host plants | Host variety |
|---|---|---|---|---|
| BCOA1 | 121 | Bird cherry-Oat aphid | *Triticum aestivum* | Chinese Spring - susceptible |
| BCOA2 [28] | 57 | Bird cherry-Oat aphid | *Triticum aestivum* | Coker 9553 (AgriPro®) - susceptible |
| CA [29] | 62 | Cannabis aphid | *Cannabis sativa* | Elite |
| | | | *Solanum tuberosum* | CO07015-4RU |
| SA [30] | 32 | Soybean aphid | *Glycine max* | PI 567301B - resistant |
| | | | | Williams 82 - susceptible |
| GPA [31] | 60 | Green Peach aphid | *Arabidopsis thaliana* | Col-0 (wild type) |
| | | | | *atg5* and *atg7* (autophagy mutants) |

of waveform samples, each with length *d*. This database was then shuffled and split into three parts: 70% for training, 10% for validation, and 20% for testing. The remaining subset (Subset 2) containing 10% the total number of recordings were reserved for the ultimate task of predicting waveform locations within complete recordings. Unlike Subset 1, recordings of Subset 2 were processed individually to maintain their completeness and avoid shuffling. Detailed data splitting information is provided in Table 2.

Next, we describe the data preprocessing techniques. In the following, we denote by $\mathbf{x} \in \mathbb{R}^{L \times d}$ a signal of *L* time steps and *d* channels.

**Normalization.** Data scaling is not mandatory but usually helps prevent numerical instability in calculations. To normalize each input recording $\mathbf{x}$, we scaled the amplitude range between $[0, 1]$ using the following formula:

$$\mathbf{x}_{\text{normalized}} = \frac{\mathbf{x} - \mathbf{x}_{max}}{\mathbf{x} - \mathbf{x}_{min}}. \tag{1}$$

**Augmentation.** Recall that in the initial segmentation stage, the input was divided into fixed-length segments *w* (default: 10.24s or 1024 time steps). Waveform samples shorter than the predefined value *d*, were padded to ensure uniform size. Doing so, we implicitly allows one padded waveform sample to contain the transition between waveforms. This is reasonable because our algorithm works by classifying each consecutive segment with fixed endpoints, e.g. from 0s to 10.23s, 10.24s to 20.47s, 20.48s to 30.71s and so on. For all samples except those of pd, a waveform sample $x = (x_i, ..., x_j)$ (where $i - j = n < d$), was extended symmetrically from its midpoint $m = \lfloor \frac{i+j}{2} \rfloor$ using the following formula:

$$x_{padded} = (x_{m-d/2+1}, ..., x_m, ..., x_{m+d/2}). \tag{2}$$

For samples of pd, whose cardinality was relatively small compare to others, we also performed *oversampling* in addition to padding to compensate for the issue. Specifically, padding was done with bigger pad size using

$$x_{padded}^{pd} = (x_{m-d+1}, ..., x_m, ..., x_{m+d}). \tag{3}$$

Then, we oversampled from $x_{padded}^{pd}$ by taking overlapping segments with a reduced step size *s* (default: 1.28s or 128 time steps), resulting in a set of $2d/s + 1$ samples for pd:

$$\text{PD\_sample} = \{(x_{m-d+1+k*s}, ..., x_{m-d+(k+1)*s}) | k \in \{0, ..., 2d/s + 1\}\}. \tag{4}$$

**Table 2. Data splitting for training ML models.** Splits of five individual datasets and a combined dataset for training, validation, and evaluating the performance of the ML models. In Subset 1, recordings were segmented, shuffled, and divided into training, validation, and testing subsets. In contrast, recordings of Subset 2 were kept intact. The combined dataset, created by concatenating all five datasets, was processed similarly for additional analysis and benchmarking.

| Dataset | Subset 1 | | | | | | | | | | Subset 2 |
|---|---|---|---|---|---|---|---|---|---|---|---|
| | Number of segments | | | Class ratios | | | | | | | n |
| | Train | Validation | Test | % NP | % C | % E1 | % E2 | % F | % G | % pd | |
| BCOA1 | 273541 | 39077 | 78154 | 0.06 | 0.12 | 0.04 | 0.57 | 0.07 | 0.02 | 0.13 | 12 |
| BCOA2 | 128733 | 18390 | 36781 | 0.22 | 0.19 | 0.08 | 0.21 | 0.0 | 0.16 | 0.14 | 6 |
| CA | 157482 | 22497 | 44994 | 0.20 | 0.25 | 0.03 | 0.14 | 0.04 | 0.11 | 0.22 | 6 |
| SA | 78762 | 11252 | 22503 | 0.31 | 0.22 | 0.01 | 0.08 | 0.04 | 0.14 | 0.2 | 3 |
| GPA | 160898 | 22985 | 45971 | 0.04 | 0.50 | 0.02 | 0.12 | 0.01 | 0.04 | 0.28 | 6 |
| Combined | 799568 | 114224 | 228448 | 0.13 | 0.24 | 0.04 | 0.29 | 0.04 | 0.08 | 0.19 | 33 |

**Features extraction.** *Fourier transform (FT)* (S1 Fig): A technique that converts signals from the time domain (amplitude/time representation) to the frequency domain (amplitude/frequency representation) by assuming that the input signal is a linear combination of sine and cosine waves at different frequencies. FT identifies which frequencies are present in the input signal by determining the coefficients of these component waves. This method has proven to be highly effective in many time series classification frameworks because it captures the characteristics of time series data well in their frequency domain representation [32–35]. Here, we fixed the hop length at 14 and the window size at 128 for generating spectrogram features.

*Wavelet transform (WT)* (S2 Fig): A technique in signal processing that assumes the input signal is a linear combination of a family of special functions called *wavelets*, which are obtained by translating and scaling a *mother wavelet*. For our study, the *Morlet* wavelet family with geometric scales of length 64 and the Symlet4 wavelet family with 3 levels of decomposition were used for the continuous WT and discrete WT, respectively. Both versions work based on the same fundamental principles, but for simplicity we focus on the discrete version. The discrete WT decomposes the original signal into several phases. In each phase, the signal is transformed into two complementary sets of wavelet coefficients: *high frequency* (detail, or "D") and *low frequency* (approximation, or "A") components. This is achieved by translating a wavelet along the time axis and convolving it with the input signal. The input signal is initially decomposed into "A1" and "D1". In each subsequent phase, the *mother wavelet* is scaled by a factor of 2, and the same decomposition process is applied to the approximation coefficients from the previous stage using this scaled wavelet. Wavelet transform has proven effective in various signal processing tasks, including analyzing human biological signals [36,37].

*Gramian Angular Field* (S3 Fig): A technique that converts a time-series into image format by representing the rescaling time-series, transforming it into polar coordinates, and finally representing it as a matrix comprising of the trigonometric sum (Gramian Angular Summation Field - GASF) or difference (Gramian Angular Difference Field - GADF) [38]. This representation allows us to apply 2D convolutional networks to time-series classification. In our study, GASF is used.

*Statistical features*: Mean, standard deviation, skewness, and quantiles are examples of these features, offering insights into data distribution. These features can capture the central tendency, dispersion, shape, and outliers of the data distribution, making them valuable for identifying changes or anomalies. Essentially, we can view these calculations as a method for

dimensionality reduction because they can describe the data through a much smaller set of numbers. The specific statistical features used in this study include mean, root mean square, standard deviation, variance, skewness, the quantiles at level 0.05, 0.25, 0.5, 0.75, 0.95, and the zero crossing rates of the consecutive segments.

*The complexity measures*: Entropy measurements provide valuable insights into a time series's regularity, long-range autocorrelation, and order structure. Shannon Entropy (SE) [39] was originally defined in classical information theory for quantifying the uncertainty of information directly through the probability of different situations. Meanwhile, Permutation Entropy (PE) [40] is a particularly robust tool for analyzing time series, as it quantifies the complexity of a dynamic system by examining the order relations between values and deriving a probability distribution of ordinal patterns. In addition to the substantial information provided, the calculation of SE and PE are relatively straightforward and quick to compute. The two entropy measurements and the statistical features are either calculated on the raw signal, the Fourier coefficients, or the discrete WT coefficients.

## Model accuracy assessment

The performance of the ML models was first assessed based on their capability to classify individual waveform samples. Since our main interest is their effectiveness in localizing the waveform, we propose an additional evaluation step using the overlap rate between the predicted and the ground-truth complete annotation. We call these (Evaluation) Task 1 and (Evaluation) Task 2, respectively. The evaluations were done using a 10-fold cross-validation scheme to avoid biases, and the mean of these metrics is reported. In the following, we explicitly present the metrics used for each evaluation task.

For a classification problem, the following metrics are commonly used:

$$\text{Accuracy} = \frac{TP + TN}{TP + TN + FP + FN},$$
$$\text{Precision} = \frac{TP}{TP + FP}, \quad \text{Recall} = \frac{TP}{TP + FP}, \quad \text{F1-score} = \frac{1}{\frac{1}{\text{Precision}} + \frac{1}{\text{Recall}}}. \tag{5}$$

More specifically, the acronyms TP, TN, FP, and FN refer to True Positive, True Negative, False Positive, and False Negative, respectively. Assume that for a classification problem, we need to distinguish among $L$ labels/classes ($L = 7$ in our case). Consider a label $L^*$; the observations/samples whose label is the same as $L^*$ are said to be Positive and Negative otherwise. True (False) Positive (Negative) refers to the number of observations that were correctly (incorrectly) classified as Positive (Negative). These quantities are often displayed in a confusion matrix of size $L \times L$ as illustrated in Table 4. The rows and columns in a confusion matrix correspond to the true and predicted labels, respectively. Hence, the diagonal elements represent correct classifications, while off-diagonal elements indicate errors.

For Task 1, as we are in a multiclass classification scenario, we decided to use the per-class accuracy scores, which measure the accuracies class-wise across the predictions on the test set. We record these values in a confusion matrix, which is normalized with respect to the number of predictions for each class. In addition, the overall accuracy (OA), which demonstrates the proportion of correct predictions across all classes, was considered. The average of the per-class F1 Score (avg-F1), which is a balanced measure of precision and recall of class-wise prediction, was also used.

For Task 2, we measure the degree of accuracy between the predicted and the manually annotated waveform. For an 8-hour recording consisting of 2,880,000 data points, 2,880,000

predicted labels are produced. The accuracy score between the predicted and the ground-truth labels (called the *overlap rate*, or OR) is calculated, showing the extent to which the predicted labels agree with the ground-truth labels.

## Experimental settings

**Setups.** For each ML model, we performed 3 groups of experiments with respect to 3 different techniques to preprocess its inputs, namely no transformation, Fourier transform, and Wavelet transform (See Table 3). Each group consists of 5 experiments on individual datasets having different settings.

One of the main goals of our study is to develop a transferable model applicable to datasets derived from various aphid species, host plants, and experimental conditions. To enhance the analysis, we further evaluated the performance of two representative models using their best input feature found in the previous experiments, 1DCNN and XGB, on the combined dataset created by merging data from BCOA1, BCOA2, CA, SA, and GPA. We selected 1DCNN to train on raw data over the other candidates because of its computational simplicity and performance comparable to the other two methods. The XGB model with WT features was selected as it is the best model in terms of both training speed and performance metrics out of the non-deep learning models.

**Models configurations.** The 1DCNN model consists of three convolutional layers that output 64, 128, and 64 channels, respectively. Each layer uses a dilation of 3 and a stride of 2, followed by ReLU activation and batch normalization. After these convolutional layers, there is a max pooling layer with a size of 3. The feature maps are then flattened, subjected to dropout with a probability of 0.5, and passed through a fully-connected layer.

The ResNet model adopts the idea of skip-connection. This model comprises three skip-connection blocks, each containing three consecutive convolutional layers with $d_1$, $d_2$, and $d_3$ output channels, respectively, followed by ReLU activation, a skip connection, and finally,

**Table 3. Input Structures of ML Models for EPG Signal Classification. Details of input structures to ML models with respect to the experimented signal processing and feature extraction techniques. In the table, $b$ is the batch size, $d$ is the predetermined length of each segment, and $n$ is the number of training observations. The abbreviations of the feature columns are None: No further feature extraction, GASF: Gramian Angular Summation Field, Spec: spectrogram, Scalo: scalogram, and Handcrafted: statistical and complexity features. For 2DCNN, all generated images are rescaled to either $64 \times 64$ or $65 \times 65$ according to the transformation technique.**

| Model | Transformation | Feature | Input shape |
|---|---|---|---|
| 1DCNN, ResNet | Raw | None | $(b, 1, d)$ |
| | FT | None | $(b, 1, d)$ |
| | WT | None | $(b, 1, d)$ |
| 2DCNN | Raw | GASF | $(b, 1, 64, 64)$ |
| | FT | Spec | $(b, 1, 65, 65)$ |
| | WT | Scalo | $(b, 1, 64, 64)$ |
| LogReg, XGB, RF | Raw | Handcrafted | $(n, 13)$ |
| | FT | Handcrafted | $(n, 13)$ |
| | WT | Handcrafted | $(n, 52)$ |

**Table 4. A confusion matrix.**

| | | Pred | |
|---|---|---|---|
| | | Positive | Negative |
| True | Positive | *TP* | *FN* |
| | Negative | *FP* | *TN* |

normalization. We set the number of output channels for each block to 64, 128, and 128. After each convolution block, max pooling with a kernel size of 3 and a stride of 3 is performed to reduce computational complexity. The outputs of the convolutional layers were flattened out, and a dropout with a probability of 0.5 was applied for normalization before being passed through a final fully connected layer.

The 2DCNN model includes two convolutional layers, each outputting 16 channels and using a $3 \times 3$ kernel with a stride of 1. Each convolutional layer is followed by ReLU activation and max pooling with a size of 2 and a stride of 2. The results are then flattened and passed through two fully connected layers with dropout to obtain the probability distribution for the 7 classes. The dropout rates are set to 0.5 for both fully-connected layers, with kernel size and stride for the max pooling layers set to 2.

The Cross-Entropy loss function is used for training the DL models, which is typical for multilabel classification tasks. The baseline hyperparameters are set to {256, 0.0001, 9, 100}, representing the batch size, learning rate, kernel size, and number of training epochs, respectively.

For traditional ML algorithms, we configured XGB with 100 trees ($n = 100$), a learning rate of 0.3 ($\eta = 0.3$), and a maximum tree depth of 6 ($d_{max} = 6$). For RF, $n$ is set to 100, but $d_{max}$ is not specified. For Logistic Regression, we used the default parameters provided by *scikit-learn*.

**Hyperparameter tuning.** Hyperparameter tuning for optimal parameter selection is critical in ML model training. However, our work only employed hyperparameter tuning on the combined dataset (i.e., on two representative models, 1DCNN and XGB) due to the high demand for computational resources. We must note that although the hyperparameter tuning process is done completely automatically, the selection of hyperparameters' ranges is heuristic, and there is no concrete procedure for this. We started with a set of heuristically chosen hyperparameters. Once their performance was validated, the values of the hyperparameters were then varied in a range including the initial value. In our study, we only perform hyperparameter tuning on the combined dataset and the XGB and 1DCNN models due to the immense computational requirements if we were to also conduct it on individual datasets.

The Grid Search (GS) technique was chosen to fine-tune the XGB model. We conducted an exhaustive search to identify the best combination by manually exploring all possible scenarios using the whole set of 52 features. The hyperparameters of interest were number of trees, learning rate, and maximum tree depth drawn from {50, 100, 200, 300}, {0.01, 0.1, 0.2, 0.3} and {3, 4, 5, 6}, respectively.

For tuning 1DCNN, we employed a more aggressive strategy due to the computational complexity, which involves manually adjusting hyperparameters in a fixed sequence: batch size, learning rate, kernel size, and then the number of epochs. The hyperparameters are selected from the following ranges: batch size {128, 256, 512, 1024, 2048, 4096}, learning rate {$10^{-2}$, $10^{-3}$, $10^{-4}$, $10^{-5}$}, kernel size {5, 9, 15}, and number of training epochs {20, 50, 100}.

**Hardware and software.** Our experiments were conducted on a personal laptop GIGA-BYTE G7 equipped with a 12-core Intel i5-10500H CPU (2.5GHz) processor and an NVIDIA GeForce RTX 3060 Laptop GPU (6GB memory). Training times ranged from half an hour to two hours, depending on the data volume, while the segmentation task was typically completed in minutes. All scripts of the DL models are implemented using the Pytorch framework [41], while the traditional ML models are implemented using *scikit-learn*. For signal processing tasks, we use *librosa* and *PyWavelet* to perform Fourier transform and Wavelet transform, respectively.

## Results

### Training on individual datasets

**Deep learning models.** The 1DCNN and ResNet models excelled and yielded superior outcomes when trained with different types of inputs, showing the strong learning capability of DL models. These models achieved near-perfect OA and average F1 scores across the three groups of experiments. The highest OAs for ResNet were 96.8%, 93.9%, and 95.3% with raw, FT, and WT inputs, respectively. Likewise, those for 1DCNN were 92.8%, 91.6% and 93.8%, respectively. On average, the OAs of ResNet were 2.6%, 2.5%, and 3.7% higher than those of 1DCNN, while ResNet also demonstrates significantly better performance in terms of the avg-F1 scores, with 4.0%, 2.6% and 6.6% higher on average. In addition, ResNet perform annotation noticeably well with simple raw input, achieving an average of 84.5% OR over the 5 datasets, with the highest of 94.8% observed in the BCOA1 dataset. However, using transformed inputs with FT and WT decreased the generalizability of ResNet by reducing the average OR of ResNet to 81.4% and 81.8%, respectively. This means that raw data were more informative to this model, as it is robust to capture relevant features from this input type. The waveforms of aphids are characterized only by certain frequency ranges. Therefore, using all Fourier coefficients for training the model might be redundant. In contrast, this additional information was useful for 1DCNN, whose architecture is much simpler than ResNet, as it increases the mean OR of 1DCNN to 80.8% using FT inputs. The simplicity in the design of 1DCNN made it less vulnerable to overfitting. Thus the more complex features were more beneficial for this model. The per-class accuracy scores given by the confusion matrix in (Fig 4) reveal that ResNet was slightly less efficient than 1DCNN when trained on a small dataset such as SA, indicating potential overfitting in the model training.

The 2DCNN model differs from 1DCNN and ResNet as it requires image inputs. To implement this model, we first generated input images of segments using three different techniques: GASF, spectrogram, and scalogram. The model yielded the best performance when trained with scalogram inputs, reaching the highest OA and avg-F1 of 95.4% and 93.5% with the BCOA1 dataset. While the spectrogram inputs may produce similar classification results, model performance with scalogram as input could achieve better generalization as there was an 8% increase in mean OR, from 75.7% with spectrogram to 83.7% with scalogram. We further investigated the confusion matrices of the 2DCNN model with scalogram input, as this approach yields the best performance. On all five datasets, we observed that 2DCNN could misclassify between pd and C waveforms, which is understandable as C is a complex waveform and pd is a part of C. The scalograms of pd may have captured redundant frequency information. With the SA dataset, 2DCNN reached perfect accuracy for E1, while ResNet and 1DCNN struggled to identify this waveform (see Fig 4). Despite experiencing fast learning after the initial 20 epochs (see Table 5), the test scores of 2DCNN show that it can generalize quite well even after 100 epochs of training.

In (Fig 5), we visualize the training process of ResNet, 1DCNN, and 2DCNN through the loss and accuracy curve when training with raw inputs. A similar pattern can be observed when training with the two other types of inputs. The training process of 1DCNN experienced a *spiking* phenomenon, a common phenomenon in stochastic gradient descent, which should only raise concern if training or validation accuracy declines. Conversely, ResNet exhibited a much smoother loss curve and a tendency to overfit rapidly, especially after just 20 epochs. This underscores the importance of robust regularization techniques to enhance its generalization capabilities across diverse datasets.

**Traditional ML models.** Regarding the non-deep learning models, the Extreme Gradient Boosting classifier (XGB) performed fairly well and achieved results that were comparable

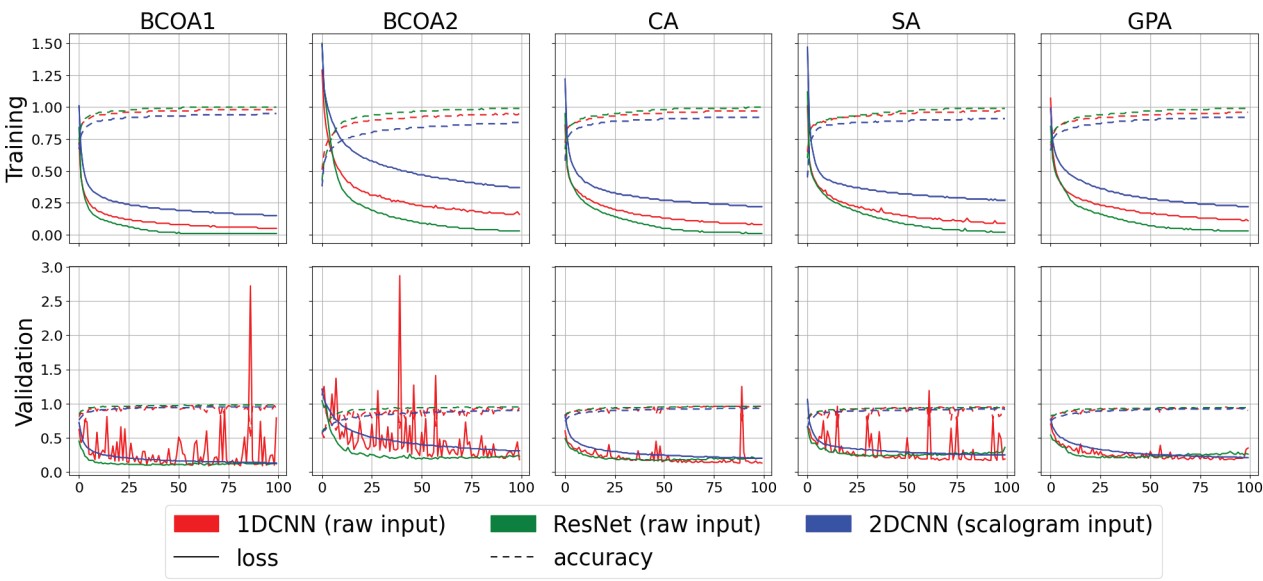

**Fig 4. Confusion matrices of deep learning models for EPG waveform classification.** The figure presents the confusion matrices for the three DL models: 1DCNN with raw inputs, ResNet with raw inputs, and 2DCNN with scalogram input features. The confusion matrices illustrate the models' performance in classifying each of the seven waveform categories (NP, C, pd, E1, E2, G, and F). The diagonal elements represent correct classifications, while off-diagonal elements indicate misclassifications.

**Table 5. Performance of DL models.** Classification and segmentation performance of 1DCNN, ResNet, and 2DCNN on 5 datasets with respect to different signal feature engineering techniques. The bold figures highlight the best mean metric of a model. Task 1 and Task 2 refers to the two evaluation tasks.

|  | Dataset | 1DCNN | | | ResNet | | | 2DCNN | | |
|---|---|---|---|---|---|---|---|---|---|---|
|  |  | Task 1 | | Task 2 | Task 1 | | Task 2 | Task 1 | | Task 2 |
|  |  | OA | Avg-F1 | OR | OA | Avg-F1 | OR | OA | Avg-F1 | OR |
| Raw | BCOA1 | 0.928 | 0.862 | 0.870 | 0.968 | 0.962 | 0.948 | 0.775 | 0.531 | 0.751 |
|  | BCOA2 | 0.914 | 0.893 | 0.647 | 0.936 | 0.907 | 0.784 | 0.688 | 0.589 | 0.532 |
|  | CA | 0.925 | 0.920 | 0.857 | 0.956 | 0.951 | 0.869 | 0.670 | 0.594 | 0.478 |
|  | SA | 0.915 | 0.839 | 0.770 | 0.914 | 0.830 | 0.769 | 0.702 | 0.528 | 0.576 |
|  | GPA | 0.904 | 0.832 | 0.732 | 0.943 | 0.894 | 0.851 | 0.840 | 0.573 | 0.704 |
| Mean |  | **0.917** | **0.869** | 0.775 | **0.943** | **0.909** | **0.844** | 0.735 | 0.563 | 0.608 |
| FT | BCOA1 | 0.916 | 0.885 | 0.913 | 0.939 | 0.914 | 0.922 | 0.947 | 0.915 | 0.860 |
|  | BCOA2 | 0.895 | 0.877 | 0.740 | 0.911 | 0.904 | 0.742 | 0.919 | 0.887 | 0.748 |
|  | CA | 0.876 | 0.870 | 0.823 | 0.922 | 0.917 | 0.847 | 0.921 | 0.916 | 0.721 |
|  | SA | 0.892 | 0.822 | 0.725 | 0.913 | 0.863 | 0.718 | 0.923 | 0.842 | 0.649 |
|  | GPA | 0.871 | 0.838 | 0.838 | 0.888 | 0.821 | 0.840 | 0.913 | 0.847 | 0.805 |
| Mean |  | 0.890 | 0.858 | **0.808** | 0.915 | 0.884 | 0.814 | 0.925 | 0.881 | 0.757 |
| WT | BCOA1 | 0.938 | 0.907 | 0.918 | 0.953 | 0.929 | 0.941 | 0.954 | 0.935 | 0.933 |
|  | BCOA2 | 0.848 | 0.795 | 0.648 | 0.922 | 0.884 | 0.702 | 0.923 | 0.896 | 0.771 |
|  | CA | 0.904 | 0.878 | 0.826 | 0.928 | 0.917 | 0.865 | 0.921 | 0.914 | 0.869 |
|  | SA | 0.910 | 0.809 | 0.692 | 0.941 | 0.876 | 0.729 | 0.913 | 0.808 | 0.755 |
|  | GPA | 0.883 | 0.754 | 0.836 | 0.927 | 0.871 | 0.855 | 0.925 | 0.876 | 0.855 |
| Mean |  | 0.897 | 0.829 | 0.784 | 0.934 | 0.895 | 0.818 | **0.927** | **0.886** | **0.837** |

to the best case of the DL algorithms, especially to the best metrics reported by ResNet, and outperformed Logistic Regression in all cases. Features extracted from WT at various resolutions demonstrated the effectiveness in characterizing the EPG waveforms, which was also the most effective of the three signal processing techniques for XGB. Indeed, WT provides the

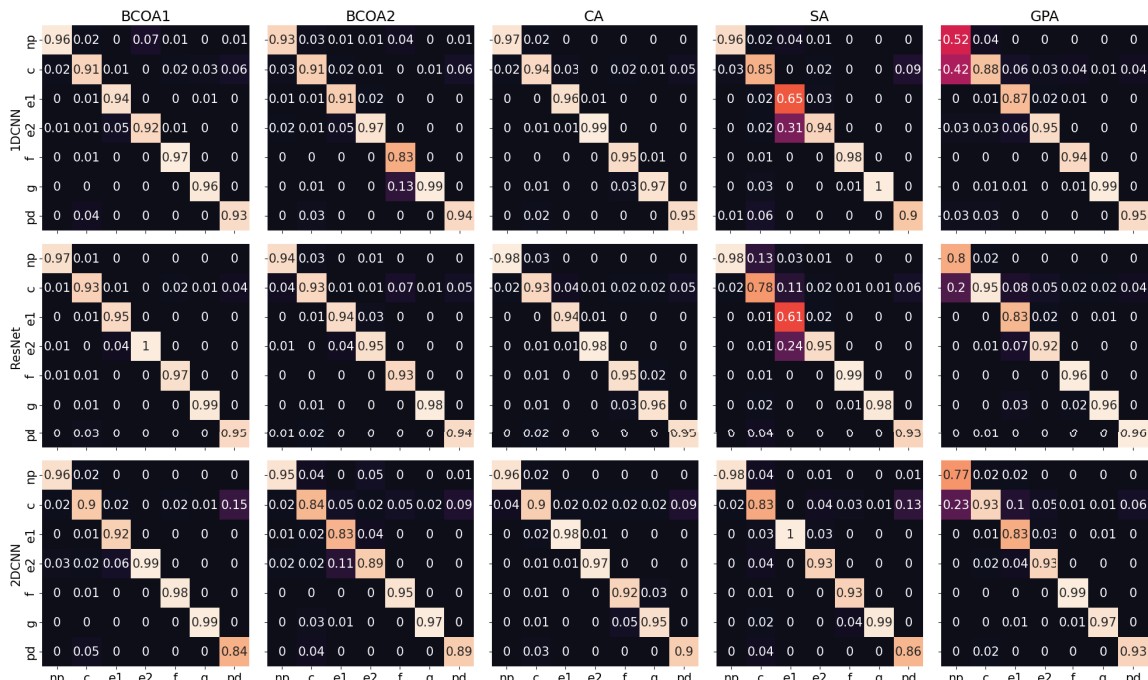

**Fig 5. Training and validation accuracy and loss over 100 epochs for deep learning models.** The graph illustrates the performance of three ML models (1DCNN, ResNet and 2DCNN) over 100 epochs of training and validation. The top and bottom panels display the measurements for the training phase and validation phase, respectively. Solid lines present the model loss (as a measure of prediction error), and dotted lines present classification accuracy (proportion of correct predictions). The distinct colors correspond to each model: red for 1DCNN, green for ResNet, and blue for 2DCNN.

models with richer information about the frequency domain of the input signal, with a total of 4 resolutions instead of 1 in FT. Moreover, the transformation principle of WT helps localize the waveform behavior regarding its frequency, which is crucial for characterizing EPG waveforms as they are defined based on the observed patterns across a certain range of frequencies. For classification tasks, the OAs range from 94.2% to 96.4%, while the avg-F1 ranges from 90.9% to 95.8%. The OR reported by XGB demonstrates a robust annotation performance, with an average OR of 84.5% across 5 datasets. Meanwhile, Random Forest (RF) with WT features showed comparative performance, and its accuracy metrics closely follow XGB's in all experiments. The highest mean OA, avg-F1, and OR metrics obtained by RF when trained on 5 datasets are 93.4%, 89.5%, and 81.8%, respectively. This is unsurprising as RF and XGB are built on the same basis of aggregating the results from a set of weaker learners - decision trees. Details of the classification results are reported in Table 6. Overall, the two models demonstrated impressive performance across global scores and individual per-class scores, often exceeding 90% with only a few exceptions that were slightly lower (see (Fig 6)).

## Training on the combined dataset

**Deep learning model.** The 1DCNN was selected out the three DL models experimented based on its competitive prediction accuracy as well as its simple architecture design. For 1DCNN, 16 experiments for hyperparameter tuning were conducted, and the outcomes are detailed in Table 7. Ultimately, we determined that a batch size of 256, a learning rate 0.0001, and a minimum kernel size 9 yield the best outcomes. The left subfigure of (Fig 7) shows the

**Table 6. Performance of traditional ML models.** Classification and segmentation performance of XGB, RF, and LogReg on 5 datasets concerning different signal feature engineering techniques. The bold figures highlight the best mean metric of a model. Task 1 and Task 2 refers to the two evaluation tasks.

| | Dataset | XGB | | | RF | | | LG | | |
|---|---|---|---|---|---|---|---|---|---|---|
| | | Task 1 | | Task 2 | Task 1 | | Task 2 | Task 1 | | Task 2 |
| | | OA | Avg-F1 | OR | OA | Avg-F1 | OR | OA | Avg-F1 | OR |
| Raw | BCOA1 | 0.911 | 0.864 | 0.776 | 0.910 | 0.864 | 0.771 | 0.616 | 0.233 | 0.604 |
| | BCOA2 | 0.945 | 0.939 | 0.602 | 0.947 | 0.942 | 0.589 | 0.541 | 0.401 | 0.555 |
| | CA | 0.907 | 0.892 | 0.704 | 0.905 | 0.891 | 0.716 | 0.628 | 0.411 | 0.585 |
| | SA | 0.915 | 0.865 | 0.631 | 0.914 | 0.870 | 0.628 | 0.706 | 0.286 | 0.644 |
| | GPA | 0.879 | 0.778 | 0.798 | 0.878 | 0.779 | 0.794 | 0.534 | 0.281 | 0.497 |
| Mean | | 0.881 | 0.867 | 0.702 | 0.911 | 0.869 | 0.699 | 0.605 | 0.322 | 0.577 |
| FT | BCOA1 | 0.886 | 0.822 | 0.841 | 0.879 | 0.813 | 0.833 | 0.673 | 0.244 | 0.653 |
| | BCOA2 | 0.915 | 0.889 | 0.579 | 0.916 | 0.885 | 0.600 | 0.454 | 0.312 | 0.593 |
| | CA | 0.907 | 0.892 | 0.705 | 0.878 | 0.862 | 0.705 | 0.587 | 0.378 | 0.580 |
| | SA | 0.894 | 0.846 | 0.616 | 0.893 | 0.841 | 0.753 | 0.632 | 0.245 | 0.685 |
| | GPA | 0.845 | 0.738 | 0.776 | 0.837 | 0.722 | 0.631 | 0.535 | 0.271 | 0.615 |
| Mean | | 0.889 | 0.837 | 0.703 | 0.881 | 0.825 | 0.705 | 0.576 | 0.29 | 0.625 |
| WT | BCOA1 | 0.964 | 0.951 | 0.932 | 0.956 | 0.940 | 0.927 | 0.776 | 0.563 | 0.756 |
| | BCOA2 | 0.962 | 0.955 | 0.789 | 0.955 | 0.952 | 0.778 | 0.691 | 0.577 | 0.709 |
| | CA | 0.957 | 0.958 | 0.856 | 0.949 | 0.949 | 0.865 | 0.760 | 0.675 | 0.692 |
| | SA | 0.951 | 0.909 | 0.787 | 0.945 | 0.892 | 0.750 | 0.757 | 0.573 | 0.730 |
| | GPA | 0.942 | 0.927 | 0.863 | 0.931 | 0.905 | 0.852 | 0.751 | 0.400 | 0.730 |
| Mean | | **0.955** | **0.940** | **0.845** | **0.947** | **0.928** | **0.835** | **0.747** | **0.558** | **0.723** |

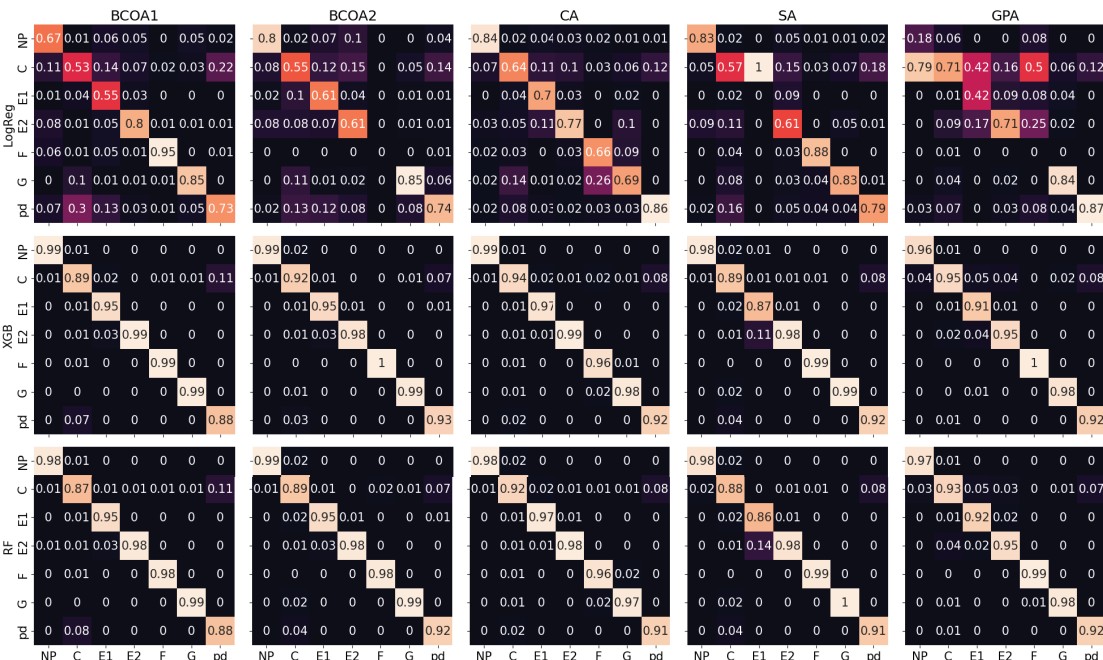

**Fig 6. Confusion matrices of traditional ML models for EPG waveform classification.** The figure presents the confusion matrices for LogReg, XGB, and RF models with features crafted from 3-level discrete WT using *symlet4* mother wavelet. As before, the confusion matrices illustrate the models' performance in classifying each waveform category (NP, C, pd, E1, E2, G, and F). The diagonal elements represent correct classifications, while off-diagonal elements indicate misclassifications.

training curves of the 1DCNN models with a batch size of 256, a learning rate of $10^{-4}$, and a kernel size of 9. As seen, a number of training epochs from 20 to 50 was generally sufficient

**Table 7. The hyperparameter tuning results of 1DCNN.** The table shows the hyperparameters tuning results of 16 experiments for 1DCNN on the combined dataset. The best model was selected based on the mean classification Accuracy (mA). We found that $\{256, 10^{-4}, 9\}$ was the best setting for batch size, learning rate, and kernel size, while 20-50 training epochs usually ensure a good learning result.

| Batch size | Learning rate | Kernel | Num. epochs | mA ↑ |
|---|---|---|---|---|
| 128 | $10^{-4}$ | 9 | 100 | 0.898 |
| 256 | $10^{-4}$ | 9 | 100 | 0.902 |
| 512 | $10^{-4}$ | 9 | 100 | 0.897 |
| 1024 | $10^{-4}$ | 9 | 100 | 0.880 |
| 2048 | $10^{-4}$ | 9 | 100 | 0.851 |
| 4096 | $10^{-4}$ | 9 | 100 | 0.821 |
| 256 | $10^{-2}$ | 9 | 100 | 0.799 |
| 256 | $10^{-3}$ | 9 | 100 | 0.837 |
| 256 | $10^{-4}$ | 9 | 100 | 0.902 |
| 256 | $10^{-5}$ | 9 | 100 | 0.847 |
| 256 | $10^{-4}$ | 5 | 100 | 0.894 |
| 256 | $10^{-4}$ | 9 | 100 | 0.902 |
| 256 | $10^{-4}$ | 15 | 100 | 0.902 |
| 256 | $10^{-4}$ | 9 | 20 | 0.894 |
| **256** | **$10^{-4}$** | **9** | **50** | **0.902** |
| 256 | $10^{-4}$ | 9 | 100 | 0.905 |

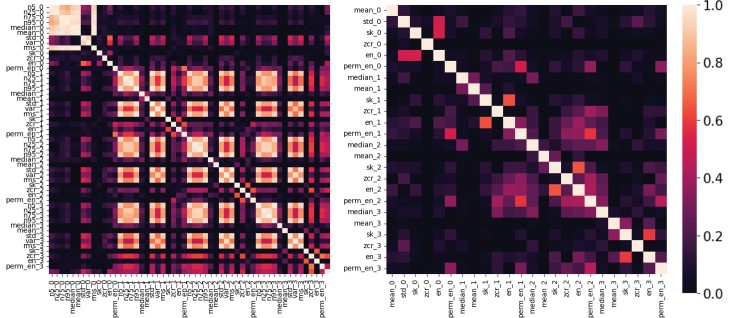

**Fig 7. Performance of 1DCNN with fine-tuned hyperparameters.** The figure illustrates the performance of 1DCNN models trained with a batch size of 256, a learning rate of $10^{-4}$, and a kernel size of 9 on EPG signal data. The left panel shows the loss (solid lines) and accuracy (dotted lines) curves during the training (blue) and validation (red) phases. These curves provide insights into the model's learning progress and ability to generalize to unseen data. The middle panel presents evaluation metrics for both tasks at epoch 100. The right panel shows the confusion matrix for the classification task, revealing the per-class accuracy and any misclassifications between waveform categories.

for the model to achieve robust learning. The middle and right figures show the final metrics for both tasks of the 1DCNN models at an epoch of 100. The 1DCNN model achieves high classification scores at 90.2% OA, and 90.2% avg-F1, respectively. The per-class accuracies showed that the highest misclassification is frequently observed between C and pd throughout our studies, primarily due to pd always being nested between the C waveform. Meanwhile, the OR rates across the 5 proposed datasets were 85.5%, 81.5%, 65.3%, 83%, and 63.1% for BCOA1, BCOA2, CA, GPA, and SA, respectively. We observe low ORs in the datasets that contain non-uniform samples of aphids' behavior, such as CA and SA, which indicates that the uniformness of the recorded EPG signal can affect the overall quality of the ML models.

**Traditional ML model.** Similar to 1DCNN, we conduct a training of a general model with Extreme Gradient Boosting Classifier (XGB) as this model demonstrate great performance in terms of accuracy and speed. For XGB, the inputs include 52 statistical and non-linear complexity features. Therefore, it is beneficial to perform feature selection (FS) as an attempt to reduce the overall computational complexity. Furthermore, FS is also pivotal to discerning correlations and determining the best training features. Employing Pearson's correlation score, we identified features with correlations exceeding 0.8 as highly correlated. Fig 8 shows the correlations matrix before and after FS, in which we note that the feature group containing the k-quantiles {0.05, 0.25, 0.5, 0.75, 0.95} at all three WT detail resolutions are highly correlated, so we only keep the median (or the 0.5 quantile) features and discard the others. Moreover, the other feature groups consisting of standard deviation, variance, and root-mean-square also show strong correlations, as they are very similar in terms of calculation. We remove variance and root mean square features and only keep the standard deviation as representative for the three features for each resolution. The remaining 24 features include the means, standard deviations, skewnesses, zero-crossing rates, and the Shannon and permutation entropies of four resolutions A3, D3, D2, and D1. We note that FS dramatically improves the feature engineering and extraction step speed, alleviating the burdensome time requirements associated with these processes.

Regarding hyperparameter tuning, the optimal hyperparameters found are {300, 0.3, 6} for a number of estimators, learning rate, and maximum depth. Table 8 shows the results of different training settings. Using a smaller set of features (FS), XGB attained a high classification accuracy for all the waveforms and only experienced a slight reduction in OA and avg-F1 metrics from 92% to 89.1% and from 92% to 88.9%, respectively. These results can be slightly improved by using the optimal set of hyperparameters obtained from Grid Search (FS + GS), which consequently yields 90.9% OA and 90.8% avg-F1. The most accurate prediction results were achieved using identical hyperparameters on the original dataset with 52 features (GS). (Fig 9) display the evaluation metrics of XGB in more detail.

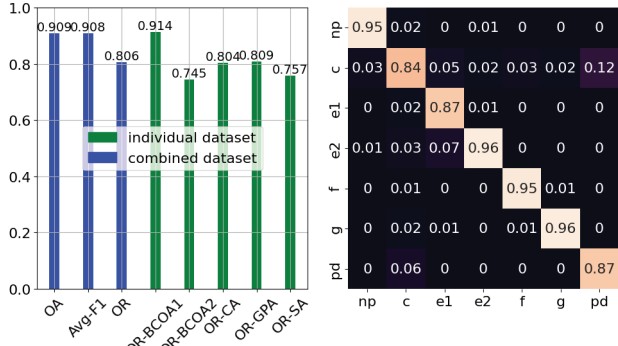

**Fig 8. Correlation matrices of handcrafted features before and after feature selection.** The figure presents two heatmaps illustrating the correlation among 52 handcrafted features derived from the 3-level wavelet transform (WT) at four resolutions. These features were used as inputs for traditional ML algorithms. The left map demonstrates the correlation matrix of 52 features crafted from 4 resolutions of the 3-level WT, which are used as inputs for traditional ML algorithms. The heat map (on the right) is the correlation matrix after feature selection, in which features whose Pearson's correlation coefficient is higher than 0.8 are considered highly correlated. We keep only one representative feature and remove its highly correlated features.

**Table 8. Features selection and hyperparameters tuning results of XGB.** Per-class accuracies, overall accuracy, and macro-average F1 score of XGB on the combined dataset with different training settings. FS: Feature Selection, GS: Grid Search.

| Techniques | Per-class accuracy | | | | | | | OA | Avg-F1 | OR |
|---|---|---|---|---|---|---|---|---|---|---|
| | NP | C | E1 | E2 | F | G | pd | | | |
| FS + GS | 0.954 | 0.844 | 0.870 | 0.956 | 0.948 | 0.960 | 0.870 | 0.909 | 0.908 | 0.808 |
| FS | 0.942 | 0.816 | 0.838 | 0.940 | 0.937 | 0.946 | 0.860 | 0.891 | 0.889 | 0.805 |
| GS | **0.978** | **0.888** | **0.924** | **0.975** | **0.972** | **0.976** | **0.888** | **0.936** | **0.937** | **0.841** |
| Baseline | 0.970 | 0.859 | 0.897 | 0.964 | 0.958 | 0.963 | 0.875 | 0.920 | 0.920 | 0.836 |

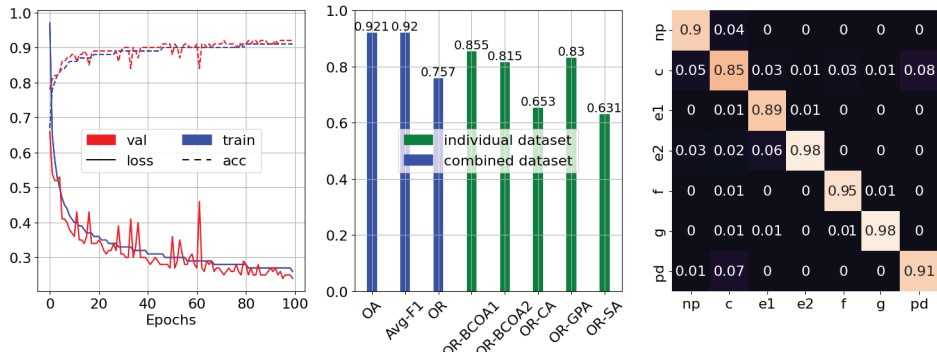

**Fig 9. Performance of XGB with fine-tuned hyperparameters.** The evaluation matrics and confusion matrix of XGB with a number of estimators of 300, the learning rate of 0.3, and a maximum depth of 6 are presented from left to right.

## Discussion

This study presents the benchmark results of six ML models in EPG signal annotation for characterizing aphid feeding behavior. The annotation procedure was implemented through three main steps: initial segmentation, waveform sample classification, and label aggregation. Two evaluation steps were conducted to thoroughly evaluate each model regarding individual waveform sample classification (Task 1) and overall annotation quality assessment (Task 2). For Task 1, the prediction accuracy of individual segments was evaluated, including the classification accuracy calculation and the F1 score. Meanwhile, the overlap rate between the predicted and the ground-truth annotated waveforms was estimated for Task 2. The results show that ML models, in general, are robust enough to annotate and characterize EPG signals and aphid-feeding behavior. Two representative models, 1DCNN and XGB, were selected based on their prediction accuracy and computational complexity to be additionally evaluated on a large aggregated dataset. The results show that our generalized models can be transferred to other datasets collected from different experimental conditions and host plant species.

Next, we shall discuss an issue regarding the waveform localization errors of our approach. In (Fig 10), the predicted annotation from the 1DCNN model almost perfectly matches the ground-truth version, reaching a commendable OR of 95%. Often, long and dominant waveforms such as G, F, and E2 are correctly identified, although instances of non-overlap are visible, which is primarily due to the offset between the predicted and ground-truth waveforms. This issue is; however, an unavoidable issue with the sliding approach since we classify the time steps every consecutive 1024 time steps, e.g., from 0 to 1024, from 1024 to 2048,

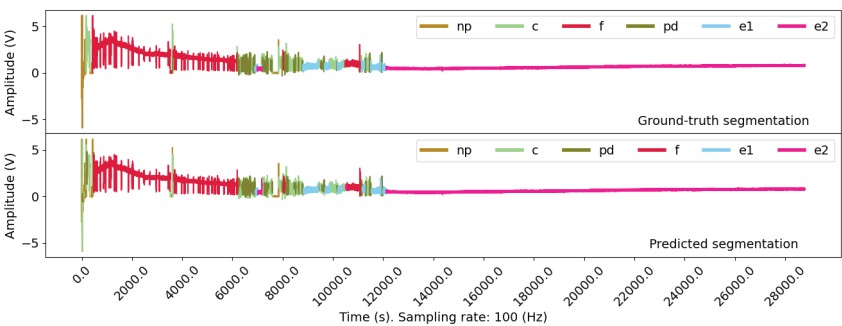

**Fig 10. Prediction of aphid behaviors from input EPG recording.** An example of ground-truth and model-predicted waveforms for a recording in the BCOA1 dataset. The behaviors of the experimented aphids are consistent, which allows ML models to capture the recurring patterns in the dataset effectively. Minor errors were noted at the endpoints of specific predicted waveforms, primarily resulting from the inherent limitations of the initial segmentation process, where recordings are divided into segments of fixed durations.

from 2048 to 3072, until we reach the end of the input recording. On the other hand, the natural behavior of aphids is not fixed in such a way, leading to the misalignment of the waveform endpoints. The phenomenon is particularly profound with short waveforms such as pd, which ML models often overlook. Thus, users should conduct a post-prediction evaluation to mitigate these inaccuracies.

Throughout the experiments, we noticed that the consistency and uniformness of aphid's behavior significantly affect the accuracy metrics used to assess the performance of the ML models, especially the OR metrics. Therefore, we present (Fig 11), which demonstrates a situation in a recording from the SA dataset where the 2DCNN model attempts to annotate a long section of the first half of the recording differently than the ground-truth version. To be explicit, the upper panel displays the ground-truth annotation containing a long green section labeled as the C waveform. In the lower panel, different labels were given to this section, which are E1, E2, and NP (in light blue, pink, and light brown, respectively). The annotation provided by the ML model in this example is reasonable, as it reflects behaviors that align with the theoretical behavior observed in other feeding stages, particularly the phloem-feeding stage. While some false positives were noted, ML models appear robust and consistent with the learned pattern. The experimental results also support this claim, as we often observe the highest accuracy metrics in experiments where aphid behaviors experience minimal variation, such as in the BCOA1 and GPA datasets. In the BCOA2 and SA datasets, where the aphid's waveform pattern is not identical throughout the experiments, the predicted waveform often has a low overlap rate with the ground-truth version. Hence, providing the ML models with well-annotated data for training is essential to ensure the best performance.

Compared to the previous works on applying ML on characterizing EPG signal, our works showed several advantages. First and foremost, our datasets is not only extensive in volume, but also diverse in biological background. It is composed of a total of 2656 hours (332 8-hour recordings) of data, compared to 470 hours [14] and only 7000s [13], while includes experiments on 4 aphid species and 5 different host plant. To our understanding, our approach is more similar to that of Xing et al. [13] in the sense that we classify waveform samples of uniform length (our default length was 10.24s compared to theirs at 10s). Regarding the classification results, the highest mean classification accuracy was achieved by ResNet at 94.43%, very close to 94.47%. Although Willett et al. [14] reported 97.4% accuracy with RF, it was difficult to make direct comparison to ours since the length and specific data processing

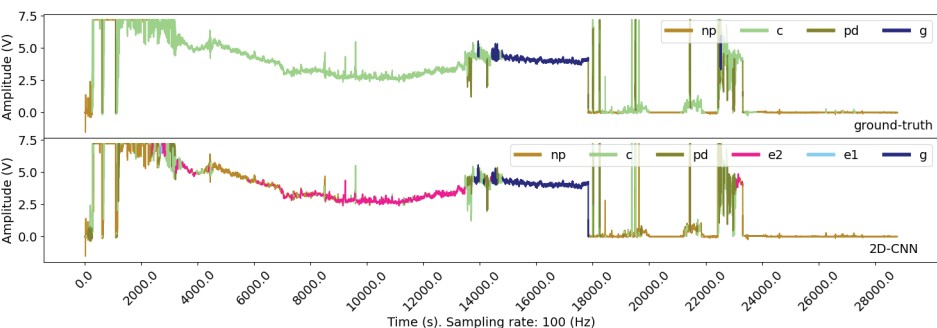

**Fig 11. Example of low overlap rate due to model misclassification in SA dataset.** The figure presents an instance where the model's predicted waveform annotation exhibits a low overlap rate with the ground-truth annotation in the SA dataset. Specifically, the misclassification of the model segment of the waveform is highlighted, demonstrating a potential area for model improvement or further investigation into the characteristics of the EPG signal in this specific dataset.

techniques were not mentioned. The advtanage of our work is also presented in the final step of our pipeline, which was the label aggregation step that returns a complete analysis file specifying the predicted locations of each waveform. Furthermore, we evaluated this aggregated result by using the OR metric, which was not done in previous work.

Nevertheless, our study demonstrates several limitations that require further investigation. As waveform samples were fixed-length and were treated *independently*, the models was not exceptionally good at annotating short waveforms such as pd that exist within C, a major waveform. The length of the waveform sample also affects the level of offset between the endpoints of the predicted and ground-truth waveform, which we are willing to tolerate. A possible solution is to adopt a more flexible algorithm that can both learn these boundary positions and classify the waveform contained within. An analogous problem to EPG signal annotation is image segmentation. Various DL models were designed to effectively tackle this problem such as Mask R-CNN, YOLO and SAM, which have greatly outperformed traditional algorithms. For time-series adaptation, U-Time [23] is a CNN-based model that has shown effectiveness for sleep-staging, a similar problem to segmenting an EPG recording. We believe that by adopting this idea, the reported results of our study can be greatly advanced.

In conclusion, we have shown that the ML methods presented can achieve great accuracy and have the potential to greatly increase the productivity of EPG research. For example, the average EPG experiment can have a sample size ranging anywhere from 15–60 recordings, depending on the number of treatments. With the average recording taking anywhere from 10-30 minutes to annotate, this can lead to hours of annotation for a given experiment. Due to the cumbersome task of manual annotation, often this task is distributed amongst multiple researchers, which can lead to potential operator error and variation in the annotated waveforms. Thus, having accurate ML tools widely available to EPG researchers will allow for higher accuracy amongst waveform calls and allow for the users to increase sample sizes without risking the addition of large analysis timelines or high operator variability. In addition, an open-source Python package available to the public will enable EPG researchers to fine-tune and customize models that can be used to better fit their research needs. For example, a research group could create a custom model based on their labs recording data for a species of interest and pick and choose which models and model parameters work best in that particular system. The availability for EPG users to have access to ML tools allows for the field of EPG to gain the rigor associated with using computational tools rather than relying solely on the

human eye. While further work needs to be done to allow even the most inexperienced users to utilize these tools, potentially via easily accessible web applications, this manuscript and the associated software packages provide a significant step in the introduction of ML tools for use in EPG waveform annotation. In conclusion, using ML tools in EPG research has the ability to enhance the accuracy of waveform classification across the field and significantly reduce the time and effort required for annotation.

## Supporting information

**S1 Fig. Fourier transform and spectrogram.** (A) An example of a 10.24s segment extracted from an E1 waveform. (B) The coefficients of positive frequency signal obtained by performing Fourier transform on the segment. This is often called the frequency domain representation of the segment. (C) Representative spectrograms of segments extracted from 7 waveforms, which are the results of short-time Fourier trasnsform.
(TIF)

**S2 Fig. Wavelet transform and scalogram.** (A) The left panel displays the hierachical structure of the discrete wavelet transform decomposition. The entire signal is initially decomposed into approximation and detail coefficient, corresponding to low and high frequency, respectively. In each subsequent stage, the approximation coefficient are further decomposed in a similar manner. On the right hand side, one observes an example of decomposing a fixed-length segment using wavelet transform with *symlet4* wavelet. (B) The scalograms of segments extracted from 7 waveforms which were obtained using *Morlet* wavelet and continuous wavelet transform.
(TIF)

**S3 Fig. Gramian angular field transformation.** (A) The process of generating a Gramian Angular Summation Field from an input time-series is presented from left to right. The time-series are transformed into polar coordinate, then convert into a matrix consisting of trigono-metrics values between pairs of angles. (B) Represntative GASFs of segments extracted from 7 waveforms.
(TIF)

## Author contributions

**Conceptualization:** Truong Son Hy, Vamsi Nalam.

**Data curation:** Vamsi Nalam.

**Formal analysis:** Quang Dung Dinh, Daniel Kunk, Truong Son Hy, Vamsi Nalam, Phuong D. Dao.

**Funding acquisition:** Vamsi Nalam, Phuong D. Dao.

**Investigation:** Quang Dung Dinh, Daniel Kunk, Truong Son Hy, Vamsi Nalam, Phuong D. Dao.

**Methodology:** Quang Dung Dinh, Daniel Kunk, Truong Son Hy, Vamsi Nalam, Phuong D. Dao.

**Project administration:** Vamsi Nalam.

**Supervision:** Truong Son Hy, Vamsi Nalam, Phuong D. Dao.

**Validation:** Quang Dung Dinh, Daniel Kunk, Phuong D. Dao.

**Visualization:** Quang Dung Dinh, Truong Son Hy.

**Writing – original draft:** Quang Dung Dinh, Daniel Kunk, Truong Son Hy.

**Writing – review & editing:** Quang Dung Dinh, Truong Son Hy, Vamsi Nalam, Phuong D. Dao.

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
