## [Decision Letter · Decision Letter 0]

10 Nov 2024

PONE-D-24-28483Machine Learning for Automated Electrical Penetration Graph Analysis of Aphid Feeding Behavior: Accelerating Research on Insect-Plant InteractionsPLOS ONE

Dear Dr. Nalam,

Thank you for submitting your manuscript to PLOS ONE. After careful consideration, we feel that it has merit but does not fully meet PLOS ONE’s publication criteria as it currently stands. Therefore, we invite you to submit a revised version of the manuscript that addresses the points raised during the review process.

Both reviews think this is valuable work. However, they also both had some concerns in addition to multiple pieces of advice about the writing and presentation. Please do your best to address all of these concerns.

We look forward to receiving your revised manuscript.

Kind regards,

Sean Michael Prager, Ph.D.

Academic Editor

PLOS ONE

Journal Requirements:

Reviewers' comments:

Reviewer's Responses to Questions

**Comments to the Author**

1. Is the manuscript technically sound, and do the data support the conclusions?

Reviewer #1: Partly

Reviewer #2: Yes

2. Has the statistical analysis been performed appropriately and rigorously? 

Reviewer #1: Yes

Reviewer #2: Yes

3. Have the authors made all data underlying the findings in their manuscript fully available?

Reviewer #1: Yes

Reviewer #2: Yes

4. Is the manuscript presented in an intelligible fashion and written in standard English?

Reviewer #1: Yes

Reviewer #2: No

5. Review Comments to the Author

Reviewer #1: The paper titled "Machine Learning for Automated Electrical Penetration Graph Analysis of Aphid Feeding Behavior: Accelerating Research on Insect-Plant Interactions" presents a promising approach to enhance the study of insect-plant interactions using advanced computational techniques. The manuscript introduces a machine learning-based method aimed at automating the analysis of electrical penetration graphs (EPGs), which could significantly accelerate research in this field. However, several aspects of the manuscript need attention to enhance its scholarly contribution and readability.

The manuscript suffers from insufficient citations, particularly in sections where foundational techniques and historical context are discussed.

The abstract could be more concise. For instance, the following sentences can be combined to enhance clarity and brevity: “This automated approach promises to accelerate research in this field significantly and has the potential to be generalized to other insect species and experimental settings. Our findings underscore the value of applying advanced computational techniques to complex biological datasets, paving the way for a more comprehensive understanding of insect-plant interactions and their broader ecological implications.”

Specific Comments:

o Historical methods can be cited appropriately: “Prior to EPG, real-time observation of aphid feeding was limited to histological methods, which only provide snapshots of the feeding process.”

o A citation is needed for the statement: “The initial EPG technique involved creating an electrical circuit by wiring the aphid and the plant, followed by recording the resulting voltage changes once the aphid closes the circuit by penetrating the plant tissue.”

o The statement “Today, the EPG technique is the predominant method for studying hemipteran feeding behavior, largely supplanting earlier histological methods” should be supported by a systematic review or study.

o Avoid lump citations such as: “Neural Networks (CNNs) have shown great promise in insect behavior characterization [19–23].”

o Do not start a sentence with a parenthesis: “(Fig 2) describes the components of each dataset.”

o In Fig. 1, consider adding pattern coding in addition to color-coding for different wave types shown in the graph.

o Line 129, Page 7: Ensure to check and provide the missing reference for: “the 2nd leaf of seedlings at Zadoks stage Z1.2 (reference to be added).”

o Table 1: Clarify what “n” stands for and provide an explanation of the variables.

o Number all equations for easy reference.

o Ensure proper citations in sections like “Gramian Angular Field” and “The complexity measures” paragraphs.

o Page 10, lines 209 to 213, can be omitted to maintain focus on the methods used and their results.

o The hyperparameter tuning method appears manual. Explain the rationale behind choosing ranges for each hyperparameter and discuss any alternative fine-tuning methods.

o Present the results of hyperparameter optimization clearly.

o Avoid describing experimental setup in the results section: “For each ML model, we performed 3 groups of experiments with respect to 3 different techniques to preprocess its inputs, namely no transformation, Fourier transform, and wavelet transform (See Table 3). Each group contained experiments on 5 individual datasets having different settings. To assess the transferability of the ML models into different scenarios, we additionally conducted experiments with two representative models from the non-deep learning and deep learning groups.”

o Define terms like TP (True Positive), TN (True Negative), etc., in equations for clarity.

o Discuss the importance of validation data and its results.

o Report the percentage of samples for each waveform.

o Evaluate the accuracy of the models based on different waveforms.

o Discuss the impact of varying sample numbers for different behaviors on the results.

o Compare the results with those of other studies to provide a broader context.

o Ensure that results for “The complexity measures” are included and discussed.

Reviewer #2: The development of a machine learning approach to processing EPG data is a critical step in removing the barriers to wider application of this technology because (as the authors point out) this is a time-consuming task and prone to human errors. The author only used aphids but used aphids under a diverse set of conditions. That diversity is a great strength to this paper. It is also impressive that these models were small and did not require some massive supercomputer. I think both the successes and “failures” are interesting and have utility for developing the next step in whatever form that will take. One take-away message for me was that the segmentation approach worked well. Further refinements of the method are worth additional exploration.

1) At the end of the introduction, I am uncertain if the goal is to analyze only aphid waveforms or if the aphid data is being used as a demonstration of how this approach will work for all insects. Will the provided program work with any data, or is retraining necessary?

a. Retraining for every aphid species would limit usefulness of the program.

b. What happens if you train on three of the aphid species and then test on the fourth?

2) It took reading the entire paper to be clear on Task 2. Somehow this should be clearer.

3) I cannot get Table 1 and Table 2 to agree. Table 1 states there were a total of 332 recordings. Each recording is 8 hours. With a segment of 10.24 seconds there are 2812.5 segments per aphid, and a total of 933750 possible segments (slightly fewer as I cannot work with a half segment). However, Table 2 has a combined total of 1142240 segments (total of train, validation, and test segments). Is this the effect of augmentation (Line 176)? Did augmentation have a different effect on different datasets?

a. In Table 2 the sum of the individual dataset for Train is less than the combined value.

4) What happens to segments that transition from one to another behavior? More what does that transition look like to the computer versus human. Do both choose the same break point?

5) There are some English language issues, both technical and in making the presentation clearer. I have caught some, but there are more. PLOS One does not edit, so I suggest trying something like the free version of Grammerly or a very careful rewrite. As is, the manuscript is readable. However, the mistakes will annoy some readers and likely cause some communication issues.

Content issues:

Author abstract) You state four aphid species on four host plants. However, the methods have four aphid species on five host plants. The host plant issue is somewhat complicated in that not each aphid is given access to all hosts. I am not bothered by that, but it makes a simple statement less accurate.

Line 11) Histology is never real-time. In histology you only know what the (now dead) insect was doing several hours (or days) in the past.

Line 19) Yes and no. The DC monitor was an improvement over the original AC monitors. However, with advances in electronics from 1964 to present we know that the AC monitor was also recording this information. However, in 1964 we had no way of uncovering this from the recorded signal. Forty year later, this is not a problem. AC or DC can be useful if one expands use past only-aphid because some insects are more sensitive to DC.

Line 21) EPG does not supplant or replace histology. Histology is ESSENTIAL to interpret the waveforms for the first time. Once the correlation is established THEN EPG can be used to monitor Hemipteran feeding behavior. If a new waveform is discovered, then histology will again be critical in understanding what that waveform means biologically.

Line 26) The gold wire is not soldered to the nail. It is glued. The gold wire breaks too easily and therefore soldering is an inefficient attachment method. Neither Freddy nor Elaine teaches soldering, though soldering would work.

Line 39) It can be shorter if the question is simple. More typically, 30 minutes grossly underestimates the time required. If my question is “Does the aphid ingest phloem” then I can scan all recordings for any presence of E2, and the task is quickly finished. If my question is “what are the differences” then I will record all behaviors, and the time required will greatly increase. Bad recordings, noise, recording time, and number of recorded behaviors all influence how long this process takes.

Line 120) Are the aphid colonies from a single mother or are they a mixture of different clones?

Line 131) This is the training data. What steps did you take to check for the human errors in validated data? Same question for all datasets. I can partly answer the question in Tutorial_EPGDataset.ipynb in that it shows that some error checking was done.

Assuming that errors were corrected, how many errors did the humans make versus the ML program?

Line 147) Was the biotype suitable to potato or might this be a laboratory rearing effect where the lab colony has largely lost any appetite for potato?

Line 176) In augmentation say (for discussion) I have a pd that lasts 4.2 seconds. I need to extend this to the left and right. How was that done? Say the 10.24 second segment has C-PD-C, I will then get three segments, the first C augmented right, the PD augmented left and right, and the second C augmented left?

Line 258) What is dynamic time-warping distance?

Line 288) Consider defining TP, TN, FP, and FN. Yes, these abbreviations are common in the ML/AI literature but not all your readers will be familiar with that literature.

Someplace you might want to describe what happens with training, validation, and testing datasets. At least a couple of sentences.

Line 291) More terms that are likely not familiar to EPG users even if many readers of PLOS One may know them.

Line 293) I do not know “w.r.t” and would there be a period after the t?

Line 296) Please define “the second task” more clearly. I will guess: In the test dataset the goal is to look at how well the model predicts new observations. However, these observations are 10.24s segments, so the “second task” is to evaluate model performance for full recordings.

Line 300) I am confused. I was picturing segments of 10.24s where there is no overlap between segments. This sounds like a moving window with the step size of the AD conversion interval (1/100 sec). In this case there would be 1024 segments containing each time step. Otherwise, there should be only one unless augmentation results in overlap.

Out of curiosity, in classifying the recordings in the second test, what is the comparison between where the human put the border between waveforms and the machine?

Line 361) reads like methods. Do you mean that all the data were used?

Table 6) What is the OR breakdown for the different datasets? This is more the average outcome, and the model trained on all the data may perform better or worse than individually trained models.

Line 515) These are not really experiments, not in the same sense as the aphid experiments.

Line 519) Equating 256 with 2^8 should not be an automatic assumption for most biologists. They can work it out, but it is not an automatic response as it should be from a computer person.

Line 525) Behaviors should never overlap. There may be cases where behaviors overlap, say when feeding and oviposition are both recorded. There may be a few other cases where overlap is present, but these behaviors should never overlap.

Figure 10) Can you make another line that is white when there is agreement and black where there is disagreement? One thing I am interested in is where disagreement takes place. Is this mostly at the transition zones, or entire behavioral events? Figure 10 suggests at least some times it is entire events.

Figure 11) Very interesting. There were more errors when a recording was not set up and the signal maxed out. It is difficult to read at this scale, but after G there are two NP spikes and then a C spike. It looks like the C spike is a ground-truth error, but I might disagree with this assessment if I could see more detail. I used the clear C at the beginning of the recording to make this assessment. Obviously, the model made some serious mistakes.

Line 528) What do you mean by “irregular samples?”

Line 553) Are you saying that I could just use the 1DCNN model and analyze my melon aphid data? Or do you mean that I could use the 1DCNN approach and train my own model where initially I would need a large data set to do the training?

Line 556) So you think the issue is short duration rather than the number of transitions? Each PD generates two transitions C-PD and PD-C.

Line 556) detailed

Line 557) Do not start a sentence with “(Fig 10)”

Line 558) matches

Line 560) waveforms

Line 586) Not so minor. I run an experiment and there are 50,000 events over the entire experiment. My error rate of 10% results in (roughly) 5000 mistakes. Depending on the nature of the mistakes it could be worth doing everything manually versus trying to find and correct that many errors. This is still a task well worth the effort, but 90% accuracy is not quite good enough to be transformative. At best, slightly useful if you are careful.

For PD, is prediction accuracy a function of PD length?

Line 585) Difficult to tell due to scale. Without exception there will be an E1 before every E2. I am having trouble distinguishing the brown of NP and the olive of PD. However, it looks like the 2D-CNN model has a NP-E2 transition at 6000 seconds.

Line 628) You are making this up. However, the overall premise is true: EPG is a labor-intensive time-consuming task prone to human error and ML is a viable solution. I think this paper shows that more conclusively than similar papers.

Edits

Line 11) The citation is hanging as a sentence all its own.

Line 13) Needs editing. Trying to combine history with method and method development. Just simplify these few sentences.

Line 16) Be clearer. Enhancing EPG equipment (AC, DC, AC-DC) is different than understanding or interpreting the biological meanings of the waveforms. That is also different from improving methods like selecting wire thickness, metal (gold versus platinum) and type of adhesives used to attach the wire to the insect. Histology does not influence the electronic development of the equipment.

Line 28) Is the electrode or the soil connected to EPG circuit?

Line 33) In manually annotating the file the user has defined the waveforms and durations. The various feeding behaviors are not manually calculated.

Line 34) The Excel workbooks calculate variables from the sequence of observed behaviors.

Line 129) Need to replace “reference to be added”. Is this “in press” or “not submitted?” Make this a valid acknowledgment of where the data came from, and then change it (if possible) as you revise the manuscript.

Line 129) datasets (you have more than one).

Line 134) In the entire manuscript remove spaces before periods at the end of sentences. It is a good idea to also use global replace to remove any double spaces generated as part of editing the document (if present).

Line 179) is “Task” here “Subset” in Table 2?

Line 187) Delete powerful.

Line 203) “converts a” and generally correct the English.

Line 257) What is TSF? This is the only use of this acronym.

Line 259) You say three and list four.

Line 260) What are LGBM, GBM, and CatBoost?

Line 261) showed

Line 262) So not all state-of-the-art performances are equal?

Figure 7) Left graph axes are difficult to read.

Line 418) What are colos?

6. PLOS authors have the option to publish the peer review history of their article (what does this mean?). If published, this will include your full peer review and any attached files.

Reviewer #1: No

Reviewer #2: No

---

## [Author Response · Author response to Decision Letter 1]

13 Jan 2025

We have addressed the reviewer's comments to the best of our ability and have included our responses in the cover letter.

---

## [Decision Letter · Decision Letter 1]

4 Feb 2025

Machine Learning for Automated Electrical Penetration Graph Analysis of Aphid Feeding Behavior: Accelerating Research on Insect-Plant Interactions

PONE-D-24-28483R1

Dear Dr. Nalam,

We’re pleased to inform you that your manuscript has been judged scientifically suitable for publication and will be formally accepted for publication once it meets all outstanding technical requirements.

Kind regards,

Sean Michael Prager, Ph.D.

Academic Editor

PLOS ONE

Additional Editor Comments (optional):

Reviewers' comments:

Reviewer's Responses to Questions

**Comments to the Author**

1. If the authors have adequately addressed your comments raised in a previous round of review and you feel that this manuscript is now acceptable for publication, you may indicate that here to bypass the “Comments to the Author” section, enter your conflict of interest statement in the “Confidential to Editor” section, and submit your "Accept" recommendation.

Reviewer #1: All comments have been addressed

Reviewer #2: (No Response)

2. Is the manuscript technically sound, and do the data support the conclusions?

Reviewer #1: (No Response)

Reviewer #2: Yes

3. Has the statistical analysis been performed appropriately and rigorously? 

Reviewer #1: (No Response)

Reviewer #2: Yes

4. Have the authors made all data underlying the findings in their manuscript fully available?

Reviewer #1: (No Response)

Reviewer #2: Yes

5. Is the manuscript presented in an intelligible fashion and written in standard English?

Reviewer #1: (No Response)

Reviewer #2: Yes

6. Review Comments to the Author

Reviewer #1: (No Response)

Reviewer #2: The paper should be published. It is a step in the right direction and such success leads to further advancement in a critical area that impedes progress in using this critical methodology. This paper clearly shows that machine learning can perform this task. It makes improvements over previous attempts and utilizes a much larger dataset than previously attempted. Overall, this is a significant step in the right direction. However, these models are not useful because the error rates are too high. Users will redirect time spent in annotating recordings into finding and correcting the errors made by the program. While this paper should be published the useful final product is still out of reach.

In a response to reviewer 2 for Line 176) It was clear that a sliding window was used, yet in the paper it seemed like a 10.24 second snapshot was used. I am not clear what parts are snapshots and what are sliding frame. Some of the following comments might be an artifact of misunderstanding this issue and could be corrected by a simple sentence somewhere in the methods.

Figure 7 is still impossible to read. If the journal publishes this as-is it will be a bad figure. My image is pixelated at a size that I could read the text. This may not be what it looks like in the journal. Consider stacking the graphs rather than side-by-side. The left hand graph is unreadable, the right hand graph is readable with difficulty. The font size on the right hand graph could be increased by a point or two to help. It looks like a label is missing on the y-axis corresponding to the row of boxes immediately above the x-axis. Maybe the journal could advise on ways to improve the figure.

Some comments could require considerable effort to address. However, please consider changing the existing text to avoid the issue rather than running more analyses.

Abstract) delete “mostly hemipterans” or make clear that you mean published literature not biological reality. Mites, thrips, mosquitoes, ticks, sand flies, ceratapogonids, and other non-hemipterans are good candidates for study using EPG. Some of these will help better understand vector-host relations in humans, livestock, and pets.

Line 33) The starting voltage value for each behavior is not used in any analysis. The voltage is used for some things, but the starting value alone is not. The Sarria workbook does not require these values. While technically correct as written for both Stylet+ and Windaq software, I would delete “, and the voltage value at the start of the behavior.”

Line 34) Variable not parameter. Parameter is used in some of the literature, but that use is a mistake and generates a unique terminology for EPG-users. Definitions from statistics or computer science predate EPG use of the term.

Line 34) By “manually” I think pencil and paper. I could see expanding this to use of a pocket calculator. I do not have an issue with the use of “manually” in line 31 (possibly because I know exactly what you mean in that case). I would discourage manual calculation. For those who choose this option I want to see where the new calculations give the same answers as the established programs. It is easy to make a programming mistake that executes without crashing the program.

Line 35) yes, that is one way to do it, but not the only way. The simple solution is to replace “Excel workbooks” with “available software.” There are programs in SAS, and PHP to name two examples of non-Excel based software used to calculate variables.

Line 55) improving (no double i)

Line 86) Host acceptance decisions occur at all stages, even NP. During NP, host acceptance is based on odor and color cues.

Line 110) In my mind a sliding window would move along the recording “millisecond by millisecond.” A snapshot approach would take successive frames that skip one or more milliseconds between frames. For this paper snapshots were taken with no overlap between successive frames.

Line 115) Why these defaults? 1024 is 2^10, but I am not clear on why this is a good default. Why not 256, or 512, or 2048?

Line 124) From this description the durations of all waveforms are to the nearest 10.24 second? So error in these models is both identifying the correct behavior and the correct duration. Short duration behaviors will be most problematic (as shown in the results).

Line 142) An RF model consists of …

Line 185) Cool, I had not considered this problem. The method will only identify pd that last more than 10.24 seconds, and more generally any event lasting less than 10.24 seconds is unlikely to be correctly identified. In your data, what is the proportion of “ground-truthed” waveform durations that are less than 10.24 seconds? How do problems in short duration events influence model accuracy?

Line 248-250) I was under the impression that the entire recording was broken into 10.24 second intervals. The intervals were then classified. However, this section suggests that (maybe) the recording was disassembled into waveforms, and each waveform was then broken into 10.24 second chunks. In the latter approach each chunk is likely to result in a 10.24 second interval at the end that is partly empty because the waveform duration does not evenly divide by 10.24 seconds. However, without explicit “data” on the transition, I fail to see how it implicitly allows for waveform transitions. A transition typically occurs quickly (<1 second). So a 10.24s frame will never catch a transition because the transition is of too short a duration to register as a unique event.

Line 299) What are zero crossing rates?

Line 314) localizing or identifying?

Line 393) What do you mean by “validated?” What was the criterion that determined “valid?”

Line 399) Best combination (of what)?

Line 400) How do I get 52 features? There are 64 combinations if number of trees (n=4), learning rate (n=4) and maximum tree depth (n=4) (4*4*4=64).

Line 422) What do you mean by different types of inputs. All the inputs are the same set of waveforms from the same set of insects for all models. There might be some variability depending on which observations are used for each of the tasks (training, validation, testing).

Line 424) near-perfect OA? I accept that the models were good, and we do not have an accuracy assessment of the humans who annotated the recordings. However, if I have 10,000 waveform events in my data and an accuracy of 93% then I have 700 mistakes. I then calculate means and variances and all of those values are tainted by 700 mistakes. This will not end well.

Line 459) There is a problem with the figures. This looks like the title for Figure 5, but I get Figure 4 in line 474.

Line 451) “best performance” relative to what or compared to what?

Line 541) “demonstrated” (use past tense as you have already done the work)

Line 541) Without rehashing the results what makes you say “…demonstrated great performance in terms of accuracy and speed?” For example, XGB in Task 2 had an OR of 0.702 versus 0.699 for RF. Is this 0.03 improvement considered “great performance?” All these processes have a certain randomness to them. Thus, the presented values all have some variance. An improvement of 0.03 seems like it might fall within this variance.

Line 557) It took at most 2 hours to train a model (line 412). That does not seem very burdensome. The 1DCNN had 6-states for batch size, 4-states for learning rate, and 3-states for each kernel size and training epochs. That yields a total of 216 possible models. At 2-hours each that would take 18 days. That might be burdensome, but in Table 7 it looks like you only examined 16 of these and that should have taken at most 1.3 days. That is hardly burdensome.

Line 594) showed

Line 595) It was good to show this, but it is also obvious that there is a decline in accuracy when presented with new data. One must ask how long will it take to sift through the data and correct the errors? Where is the break even point where the time it takes to correct all the errors is more than how long it would have taken manually. As an alternative, how reliable are standard statistical methods (ANOVA, regression, and so forth) when one knows that 2 to 5% of the values are wrong? Accepting this is great in an era of fake news and alternative facts, but this is not the science that I had to learn.

Line 598) “commendable” only in the broader context of AI/ML model accuracy. However, the standards are different when the AI is generating data that will be used in additional analyses. To be useful you need to be closer to 100% accuracy (in my opinion).

Line 608) The placement of Figure 10 appears to be off. I can accept tables and figures in the discussion, but they should reinterpret the data or show relationships with previously published data. They can help interpret results, not present new results.

Line 632) this is providing a discussion of difficulties in using one model for all aphids and a caution for using one model for all conditions. Individuals from high stress treatments or electronically noisy laboratory environments may be problematic. Some discussion of the point where the model must be retrained would be useful. If the answer is that the model needs retraining for every experiment, then machine learning is useless for this task.

Line 634) More results are in the Discussion.

Paragraph starting line 640) Correct the English, otherwise a good addition.

Line 652) Please revise the entire manuscript in this regard: “Location” is not something I care about. The data that I need consists entirely of waveform name and a duration. There are alternatives like “Time to start of event” or “time to end of event” could be interpreted as “location” but stating specifically start or end of event is more informative. If I get back a file with a waveform for each 10-millisecond interval I will have to run additional processing steps. Accuracy for me has two parts: 1) Was the waveform correctly identified and 2) Is the duration of each event correct.

Line 652) If I am thinking about using this program in my research, then this is the most important step in the entire paper. This is the output that I will need in analyzing my data. As scientist and reviewer, I am also interested in the errors made by the machine learning program, and how those errors were corrected. I accept that there are problems with pd, but other behaviors can also last <10 seconds. What part of model accuracy is a difficult to identify waveform and what part is a difficulty in waveform duration versus frame size?

(Something to consider – not a change!) There are two missing (less common) waveforms: E1e and E1E2. Might some of the error rate be a problem in these waveforms? E1E2 could cause problems because it is an alternation between E1 and E2 behaviors and typically the duration of each E1 or E2 segment is less than 10.24 seconds.

Line 665) Waveform samples were of fixed length: I can see this as a problem, but I have no solution. Backus has suggested a 2-second rule. If a behavior lasts <2 seconds then it is ignored. First, can you see a pattern in a behavior lasting <2 seconds? Second, this rule prevents every cluster of random spikes in NP being classified as C.

Line 665) Independence: This needs more explanation. I do not think that independence in the statistical sense is relevant for ML models. At least in image analysis I can take the exact same picture but rotate it 33-degrees and get a new image that is perfectly valid for training models. It is more about the degree of correlation between two image matrices rather than some notion of independence.

Line 659) delete “which we are willing to tolerate” or add more context. The offset generates errors in the duration data, and we will have to decide if that error is tolerable.

Line 661) So training needs to have these snapshots. That does not mean that the analysis phase must use the same approach. Yes, the frame size also must be 10.24 seconds, but I can take a 10.24 second frame and move it in 10 milliseconds increments for an entire recording. The result for every 10 ms recording is the outcome of the analysis of the 10.24 ms following that point. That will give you a better resolution and fewer errors in Task 2.

Line 670) The recommended is 20 recordings per treatment. People have published with fewer than ten. The number of recordings is the upper limit to the number of replicates. Not all individuals display all behaviors.

Line 672) My recordings last 24 hours because the insect takes that long to start ingestion. Long recordings may also be required for insects that spend a long-time ingesting. A paper on nymphal behavior recorded 48 h because the larvae would not stop. Ticks, and more sessile phytophagous insects are also problems. The recording length must match the insect behavior to get meaningful results. It would be wonderful if I could annotate an entire experiment in only a few hours.

Line 674) I keep the task to one researcher for any given experiment. There is researcher-to-researcher variability that I would not want to add into my data (as you point out would be the case).

Line 685 to end) I agree!

Reviewer comments and responses:

1) I am unclear on how the hyperparameters were chosen as both starting value or as a range. Many readers will not care, so might this be in a supplement? One of the issues is understanding model performance outside the range but also issues with local minima/maxima. Some models improve if hyperparameters change once the model stops improving epoch-over-epoch.

2) The imbalance in the number of classes is not “unavoidable.” Take the class with the fewest number of observations and then take a random sample of that size from the other classes before training the model. A simple programming step resolves the issue but can result in more variability in model metrics depending on which images are selected in the random process. Alternatively, a human could select specific images in a way to achieve class balance.

7. PLOS authors have the option to publish the peer review history of their article (what does this mean?). If published, this will include your full peer review and any attached files.

Reviewer #1: No

Reviewer #2: No

---

## [Editor Report · Acceptance letter]

PONE-D-24-28483R1

PLOS ONE

Dear Dr. Vamsi,

I'm pleased to inform you that your manuscript has been deemed suitable for publication in PLOS ONE. Congratulations! Your manuscript is now being handed over to our production team.

Kind regards,

on behalf of

Dr. Sean Michael Prager

Academic Editor

PLOS ONE